# Importance of considering riparian vegetation requirements for the long-term efficiency of environmental flows on aquatic microhabitat

Rui Rivaes[1], Isabel Boavida[2], José M. Santos[1], António N. Pinheiro[2], Teresa Ferreira[1]

[1]Forest Research Centre, Instituto Superior de Agronomia, Universidade de Lisboa, Tapada da Ajuda 1349-017 Lisboa, Portugal
[2]CERIS, Civil Engineering Research Innovation and Sustainability Centre, Instituto Superior Técnico, Universidade de Lisboa, Av. Rovisco Pais, 1049-001 Lisboa, Portugal

*Correspondence:* Rui Rivaes (ruirivaes@isa.ulisboa.pt)

**Abstract.** Environmental flows remain biased toward the traditional biological group of fish species. Consequently, these flows ignore the inter-annual flow variability that rules species with longer life cycles and therefore disregard the long-term perspective of the riverine ecosystem. We analyzed the importance of considering riparian requirements for the long-term efficiency of environmental flows. For that analysis, we modeled the riparian vegetation development for a decade facing different environmental flows in two case studies. Next, we assessed the corresponding fish habitat availability of three common fish species in each of the resulting riparian landscape scenarios. Modeling results demonstrated that the environmental flows disregarding riparian vegetation requirements promoted riparian degradation, particularly vegetation encroachment. Such circumstance altered the hydraulic characteristics of the river channel where flow depths and velocities underwent local changes up to 10 cm and 40 cm s$^{-1}$, respectively. Accordingly, after a decade of this flow regime, the available habitat area for the considered fish species experienced modifications up to 110% when compared to the natural habitat. In turn, environmental flows regarding riparian vegetation requirements were able to maintain riparian vegetation near natural standards, thereby preserving the hydraulic characteristics of the river channel and sustaining the fish habitat close to the natural condition. As a result, fish habitat availability never changed more than 17% from the natural habitat.

## 1 Introduction

Freshwater ecosystems provide vital services for human existence but are on top of the world's most threatened ecosystems (Dudgeon et al., 2006; Revenga et al., 2000), primarily due to river damming (Allan and Castillo, 2007). The ability to provide sufficient water to ensure the functioning of freshwater ecosystems is an important concern as its capacity to provide goods and services is sustained by water-dependent ecological processes (Acreman, 2001). The relevance of this subject compelled the scientific community to appeal to all governments and water-related institutions across the globe to engage in environmental flow restoration and maintenance in every river (Brisbane Declaration, 2007). Actually, this issue is a global reach topic, as all dams, weirs and levees change the magnitude of peak flood flows of rivers to a certain extent (e.g., FitzHugh and Vogel, 2010; Maheshwari et al., 1995; Miller et al., 2013; Nilsson and Berggren, 2000; Uddin et al., 2014a, b). As a result of this, there are still opportunities for the

implementation of environmental flow restoration at hundreds of thousands of these structures worldwide (Richter and Thomas, 2007).

Environmental flows can be defined as "the quantity, timing and quality of water flows required to sustain freshwater and estuarine ecosystems, and the human livelihoods and wellbeing that depend upon these ecosystems" (Brisbane Declaration, 2007) and play an essential role in the conservation of freshwater ecosystems (Arthington et al., 2006; Hughes and Rood, 2003). It is now in agreement that environmental flows must ideally be based on the ecological requirements of different biological communities (e.g., Acreman et al., 2009; Acreman and Ferguson, 2010; Acreman et al., 2014; Arthington et al., 2010; Arthington, 2012; Arthington and Zalucki, 1998; Davis and Hirji, 2003; Dyson et al., 2003; Poff et al., 1997) and should present a dynamic and variable hydrological regime to maintain the native biodiversity and the ecological processes that portray every river (Bunn and Arthington, 2002; Lytle and Poff, 2004; Postel and Richter, 2003). In this sense, holistic methodologies meant to address river systems as a whole (Arthington et al., 1992; King and Tharme, 1994; King and Louw, 1998) are clearly being increasingly applied out of Australia and South Africa (Hirji and Davis, 2009), the origin countries of this holistic concept. However, the most commonly applied methods throughout the world are still hydrologically based methods (Dyson et al., 2003; Linnansaari et al., 2012; Tharme, 2003). Conversely, environmental flows ascertained through habitat simulation methods still persist generally based on the requirements of a single biological group, mostly fish (Acreman et al., 2009; Arthington, 2012; Tharme, 2003), and require an input from less typically monitored taxa (Gillespie et al., 2014). Accordingly, these approaches still disregard the inter-annual flow variability that rules species with longer lifecycles, like riparian vegetation, therefore lacking the long-term perspective of the riverine ecosystem (Stromberg et al., 2010). The feedbacks of these shortcomings on the riparian and aquatic communities were seldom estimated before and so, the efficiency of such approaches along with its long-term after-effects remains practically unknown.

Riparian vegetation is a suitable environmental change indicator (Benjankar et al., 2012; Nilsson and Berggren, 2000) that responds directly to flow regime in an inter-annual timeframe (Capon and Dowe, 2007; Naiman et al., 2005; Poff et al., 1997) and has a clear significance in the habitat improvement of aquatic systems (e.g., Broadmeadow and Nisbet, 2004; Chase et al., 2016; Dosskey et al., 2010; Gregory et al., 1991; Pusey and Arthington, 2003; Rood et al., 2015; Ryan et al., 2013; Salemi et al., 2012; Statzner, 2012; Tabacchi et al., 2000; Van Looy et al., 2013; Wootton, 2012). In fact, riparian vegetation and aquatic species interact biologically, physically and chemically (Gregory et al., 1991). Riparian vegetation is capable of influencing aquatic species in several ways. It affects food webs by providing an important input of nutrients that are a major food source for invertebrates, which are in turn eaten by fishes (Wootton, 2012). It influences hydrological processes (Salemi et al., 2012; Tabacchi et al., 2000) and protects aquatic habitats by means of river bank stability (Rood et al., 2015) and providence of large woody debris (Fetherston et al., 1995). It provides thermal regulation of rivers by overshadowing (Ryan et al., 2013) and protect water quality both by trapping sediments and contaminants (Chase et al., 2016) as by chemical uptake and cycling (Dosskey et al., 2010). On the other hand, aquatic species appear also to be able to influence riparian zones, although in a much smaller magnitude, acting as ecosystem engineers (Statzner, 2012). For instance, fishes can dig in sand and gravel for food or reproductive purposes and therefore

influence sediment surface characteristics and critical shear stress (e.g., Hassan et al., 2008; Statzner et al., 2003).

Accordingly, riparian restoration is an indispensable implementation measure to recover the natural river processes and is the most promising restoration action in many degraded rivers (Palmer et al., 2014). Hence, incorporating riparian vegetation requirements (the need for specific flows to preserve the naturalness of recruitment and meta-stability facing fluvial processes) into environmental flows could be an important contribution to fill in these gaps.

We have already noticed how environmental flow regimes disregarding riparian vegetation requirements allow for the degradation of riparian woodlands in the subsequent years following such river regulation (e.g., Rivaes et al., 2015). However, we are not aware of studies assessing the comeback of this degradation again on the efficiency of those environmental flow regimes. The purpose of this study is to evaluate the effect of disregarding riparian vegetation requirements in the efficiency of environmental flow regimes regarding fish habitat availability in the long-term perspective of the fluvial ecosystem. We used an approach from an ecohydraulic point of view to evaluate the effects of riparian landscape degradation on fish species. By riparian landscape we mean the specific spatial patterns of riparian vegetation that result from ecological, geomorphological and hydrological processes, and are depicted by the existing patch mosaic with different vegetation types and succession phases. We were particularly interested in answering the following questions: i) are environmental flows exclusively addressing fish requirements capable of preserving the habitat availability of these aquatic species in the long-term? ii) If not, to what extent can the disregard for riparian vegetation requirements derail the goals of environmental flows addressing only aquatic species as a result of the riparian landscape degradation? iii) Are environmental flows regarding riparian requirements able to maintain the habitat availability of fish species?

To approach these questions, we first modeled the structural response of riparian vegetation (please see Naiman et al., 2005 and NRC, 2002 for a better understanding about riparian vegetation structure) facing a decade of different environmental flows in two different case studies. Next, we performed an assessment of habitat availability for fish species in each of the resulting riparian landscape scenarios. We are not aware of such a modeling approach ever being used in the appraisal of the long-term efficiency of environmental flow regimes, which can provide an extremely valuable insight of the expected long-term effects of environmental flows in river ecosystems in advance.

## 2 Methods

### 2.1 Study sites

The two study sites were selected in the Ocreza River, East Portugal (Figure 1). This is a medium-sized stream that runs on schistose rocks for 94 km and drains a 1429 $km^2$ watershed with a mean annual flow of 16.5 $m^3$ $s^{-1}$. The flow regime is typically Mediterranean (Gasith and Resh, 1999), with a low flow period interrupted by flash floods in winter (median of mean daily discharges in the winter months is 8.8 $m^3$ $s^{-1}$ and maximum annual discharges with a return period of 2, 5, 10 and 100 years are respectively 323, 549, 718 and 1314 $m^3$ $s^{-1}$) and a very low flow, even null at times, during summer (the first quartile and median of mean daily discharges in summer months is respectively 0 and 0.1 $m^3$ $s^{-1}$). Two study sites were

considered (OCBA and OCPR) to provide a broader analysis of the aquatic habitat modifications in different hydrogeomorphological contexts. The OCBA study site (39° 44' 07.05" N, 7° 44' 16.51" W) is located 30 km upstream from the river mouth and OCPR (39° 43' 16.88" N, 7° 46' 01.05" W) is approximately 5 km downstream of OCBA. Despite the relatively small distance between them, several

characteristics differentiate the two study sites. While in OCBA, the river flows freely on a boulder substrate and is confined to steep valley hillsides, in OCPR, the river flows on a coarser boulder substrate with sparse bedrock presence and is located in a relatively wider valley section. OCBA and OCPR also contrast in watershed areas, representing 54 and 72% of the entire river basin, respectively. This feature further differentiates the two case studies, as the intermediate watershed of OCPR collects water from a much

rainier zone, therefore conferring an increased flow regime in this study site. The surveyed areas in the OCBA and OCPR study sites encompass a river length of approximately 500 and 300 m, respectively, laterally limited by the 100-year flooded zone, thus totaling approximately 4 and 3 ha for OCBA and OCPR study sites respectively. In both cases, the fish community is characterized by native cyprinid species, mainly *Luciobarbus bocagei* (Iberian barbel, hereafter barbel), *Pseudochondrostoma polylepis* (Iberian

straight-mouth nase, hereafter nase) and *Squalius alburnoides* (calandino), whereas the local riparian vegetation is composed mostly of willows (*Salix salviifolia* Brot. and *Salix atrocinerea* Brot.) and ashes (*Fraxinus angustifolia* Vahl).

### 2.2 Data collection

#### 2.2.1 Hydraulic data

The riverbed topography was surveyed in 2013 using a combination of a Nikon DTM330 total station and a Global Positioning System (GPS) (Ashtech, model Pro Mark2). Altogether, 7707 points were surveyed at OCBA and 25132 at OCPR. Trees, boulders and large objects emerging from the water were defined by marking the object intersection with the riverbed and by surveying the points necessary to approximately define its shape.

Hydraulic data –, i.e., water velocities and depths – were measured as a series of points along several cross-sections in the study sites. Depths were measured with a ruler and water velocities with a flow probe (model 002, Valeport) positioned at 60% of the local depth below the surface (Bovee and Milhous, 1978). Additionally, the substrate composition was visually assessed and mapped to determine posteriorly the effective roughness heights of the riverbed. These data were used to calculate river discharge in each study

site and to calibrate the model. Additional information about hydraulic data and channel bed characteristics is provided as supplementary material (Appendix A – Tables A1, A2, A3 and A4).

#### 2.2.2 Riparian vegetation data

The riparian vegetation was assessed in 2013 to support the calibration and validation of the riparian vegetation model. This task consisted in recording the location and shape of all homogeneous vegetation

patches with a sub-meter precision handheld GPS (Ashtech, Mobile Mapper 100), while dendrochronological methods were used to determine the approximate age of the patches. Two or three of the largest individuals in each patch were cored with a standard 5 mm increment borer, taking two perpendicular cores at breast height in adult trees (Mäkinen and Vanninen, 1999). For individualswith a

diameter smaller than 5 cm at breast height, discs were obtained for age calculation purposes, and on multistemmed trees, the cores/discs were taken from the largest stem. The patches were later classified by succession phase according to its corresponding development stage. Patch georeferencing, patch aging and succession phase classification followed the methodology used by Rivaes et al. (2013).

Five succession phases were identified in the study sites: Initial phase (IP), Pioneer phase (PP), Early Successional Woodland phase (ES), Established Forest phase (EF), and Mature Forest phase (MF). Initial phase was attributed to all patches dominated by gravel bars, sometimes covered by herbaceous vegetation but without woody arboreal species. The patches dominated by the recruitment of woody arboreal species were considered as Pioneer phase. The Early Successional Woodland phase classification was attributed to all patches with a high standing biomass and well-established individuals, dominated by pioneer watertable-dependent species, such as willows and alders (*Alnus glutinosa*). Older patches dominated by macrophanerophytes, such as ash-trees, were considered to be in Established Forest phase. The Mature Forest phase was considered at patches where terrestrial vegetation was also present, determining the transition phase to the upland vegetation communities. Further information on the characterization of succession phases is provided as supplementary material (Appendix B – Table B1 and Figures B1 and B2).

### 2.2.3 Fish data

Fish populations were sampled during 2012 and 2013 at undisturbed or minimally disturbed sites in the Ocreza basin, an essential requisite when studying habitat preferences of stream fishes in order to reflect their optimal habitat (Gorman and Karr, 1978). Sampling occurred in autumn (November, 2012), spring (May, 2013) and early summer (June, 2013) when there is full connectivity among instream habitats. Overall, four native species (cyprinids) were found – barbel, nase, calandino and the Southern Iberian chub (*Squalius pyrenaicus*). The latter was however excluded from the present study, as an insufficient number of individuals were collected to draw unbiased conclusions. Non-native fish (the gudgeon *Gobio lozanoi*) occurred in the study area, but in very low density. Field procedures followed those by Boavida et al. (2011, 2015). Fish sampling was performed during daylight using pulsed DC electrofishing (SAREL model WFC7-HV; Electracatch International, Wolverhampton, UK), with low voltage (250 V) and a 30 cm diameter anode to reduce the effect of positive galvanotaxis. A 200 m long reach at each site was surveyed by wading upstream in a zigzag pattern to ensure full coverage of available habitats. To avoid displacements of individuals from their original positions, a modified point electrofishing procedure was employed (Copp, 1989). Sampling points were approached discreetly, and the activated anode was swiftly immersed in the water for five seconds. Upon sighting a fish or a shoal of fishes, a numbered location marker was anchored to the streambed for subsequent microhabitat use measurements. Fish were immediately collected by means of a separate dip net held by another operator, quickly measured for total length (TL), and then placed in buckets with portable ELITE aerators to avoid continuous shocking and repeated counting, before being returned alive to the river. Ensuing fish sampling, microhabitat measurements of flow depth (cm), mean water velocity (cm s$^{-1}$) and dominant substrate composition were taken in 0.8 by 0.8 m quadrats at the location where each fish was captured. Microhabitat availability measurements were made using the same variables by quantifying randomly selected points along 15–25 m equidistant transects perpendicular to the flow at each sampling site. To develop Habitat Suitability Curves (HSC) for target fish size classes,

microhabitat variables (flow depth, water velocity, dominant substrate and cover) were divided into classes, and histograms of frequencies of use and availability were constructed (Boavida et al., 2011). A summary on collected fish data, as well as data analysis to determine habitat use, availability and preference of fish species regarding the considered variables, is provided as supplementary material (Appendix B – Table B2 and Figures B3 to B12).

## 2.3 Flow regime definition

Three flow regimes were considered for the modeling of riparian vegetation: i) the natural flow regime (hereafter named natural flow regime), ii) an environmental flow regime considering only fish requirements (hereafter named Eflow regime) and iii) an environmental flow regime considering both fish and riparian requirements (hereafter named Eflow&Flush regime). The natural flow regime data was obtained from the Portuguese Water Resources National Information System (SNIRH, 2010). The environmental flow regimes used in this study are an adaptation from the environmental flow regime created by Ferreira et al. (2014) for the location of the study sites (Figure 2). These authors determined an environmental flow regime presented in a multiannual fashion considering a decadal time frame and accounting for two different flow regime components: a monthly flow regime addressing fish requirements and a multiannual flow regime composed by floods with different recurrence intervals addressing riparian vegetation requirements. The first component, i.e., the flow regime addressing fish requirements (Eflow), was determined according to the Instream Flow Incremental Methodology (Bovee, 1982) and was built on a monthly basis to embody the intra-annual variability ruling the main life cycle events of this biological group (Encina et al., 2006; Gasith and Resh, 1999). These mean monthly discharges addressing fish requirements that compose the Eflow aimed for the following goals: i) maximize the habitat of the target species while attributing the same weight for each species; ii) privilege the spawning months (spring; Santos et al., 2005) and promote the younger life stages during summer; iii) maintain the characteristic intra-annual variability of the river flow; and iv) preserve the natural regime whenever the environmental flows suggest higher discharges. The second component of the environmental flow regime (floods with a certain recurrence interval) proposed by Ferreira et al. (2014) was determined according to Rivaes et al. (2015) and intend to characterize the inter-annual flow variability to which the arrangement of riparian vegetation communities respond (Hughes, 1997). The flushing flows addressing riparian requirements in the Eflow&Flush regime were defined based on the need of riparian communities for the minimum necessary flushing flow regime to maintain the viability and sustainability of riparian vegetation, particularly, avoiding vegetation encroachment and conserving the ecological succession equilibrium of the riparian ecosystem (Rivaes et al., 2015). Therefore, the environmental flow regimes used in this study are considered an adaptation from Ferreira et al. (2014) as we used just the fish-addressing component (only mean monthly discharges) as the standard procedure of an environmental flow regime considering only fish requirements (Eflow) and both components (mean monthly discharges and flushing flows) for the environmental flow regime addressing fish and riparian requirements (Eflow&Flush).

## 2.4 Riparian vegetation modeling

The riparian vegetation modeling was performed using the *CASiMiR-vegetation* model (Benjankar et al., 2009). This tool simulates the succession dynamics of riparian vegetation, based on the existing relationships of the ecological relevant hydrological elements (Poff et al., 1997) and the vegetation metrics that reflect riparian communities to such hydrological alterations (Merritt et al., 2010). The strengths of this model are the capacity of incorporating the past patch dynamics into every model run, the ability of working at a response guild level by using succession phases as modeling units, and the ability of providing the outputs in a spatially-explicit way. In turn, main disadvantages of this model can be attributed to the inexistence of a plant competition module or the lack of an incorporated hydrodynamic model.

The rational of this model is based on the fact that riparian communities respond to the hydrological and habitat variations on a time scale between the year and the decade (Frissell et al., 1986; Thorp et al., 2008), being that the flood pulse is the predominant factor on these population dynamics (Thoms and Parsons, 2002). For these reasons, the hydrological regime is inputted into the model in terms of maximum annual discharges as these discharges are considered as the annual threshold for riparian morphodynamic disturbance that determine the succession or retrogression of vegetation. Notwithstanding, the model also predicts the annual riparian adjustments according to its vital rates in relation to groundwater depth, as well as the annual recruitment areas, based on the annual minimum mean daily discharges. The groundwater depth corresponding to the mean annual discharge of the river is also a model input used as a reference for the general habitat conditions that determine the expected riparian landscape according to the calibrated thresholds of the riparian succession phases. Thus, the magnitude and duration of extreme low flows are accounted by CASiMiR-vegetation model. A complete detailing of model rational and parameterization can be found in Politti and Egger (2011) and Benjankar et al. (2011). Model calibration was carried out in accordance with the methodology described in previous studies (García-Arias et al., 2013; Rivaes et al., 2013). Particularly, calibration was performed by running the CASiMiR-vegetation model for a decade to simulate the effect of the local historic flow regime on riparian vegetation. The result of the model was then compared with an observed vegetation map that was surveyed in the same year of the one corresponding to the result of the model. This is an iterative process of trial an error where the parameter of shear stress resistance threshold of each succession phase is tuned to obtain the best calibration outcome (see Wainwright and Mulligan, 2004, for a better understanding). All the other parameters, namely, patch age and height above water table ranges were determined based on the data collected in the field. This information is provided as supplementary material (Appendix A – Table A5). During calibration, the riparian vegetation model achieved an agreement evaluation of 0.61 by the quadratic weighted kappa (Cohen, 1960), which is considered to be in good agreement with the observed riparian landscape (Altman, 1991; Viera and Garrett, 2005). This agreement evaluation can be understood as a classification 61% better than what would be expected by a random assignment of classes. The riparian vegetation model was further validated in this specific watershed (Ferreira et al., 2014) with even better results (quadratic weighted kappa of 0.68). After calibration and validation (calibrated parameters provided as supplementary material; Appendix A – Table A5), the riparian vegetation was modeled for periods of ten years according to the corresponding flow regimes (Table 1). Such modeling period was considered to be long enough to avoid the influence of the initial vegetation conditions, while river morphological changes still do not assume

importance in vegetation development (Politti et al., 2014). Furthermore, during modeling, riverbed topography was considered fixed for several reasons: the study sites are located in a fairly steep valley in which river is not allowed to meander considerably during such a short time scale; the typical substrate of both study sites is armored and very coarse (boulders, large boulders and bedrock); in these conditions the

small monthly discharges intended to maintain aquatic fauna requirements are not able to create water depths and flow velocities capable of moving or eroding particles with the size of those found as substrate in the considered study sites (for a better understanding please see Alexander and Cooker, 2016; Clarke and Hansen, 1996; Hjulström, 1939); no significant differences were found during the substrate analysis of the different succession phases; prior knowledge of the authors show that the considered floods do not bring

noteworthy changes to river geomorphology during this period (Rivaes et al., 2015); the model calibration and validation results exhibited a good agreement with the observed riparian landscape while using the same methodology; by using a fixed topography it is possible to analyze the exclusive effect of riparian landscape degradation on the river hydraulics.

The resulting riparian vegetation maps were then used as the respective riparian landscapes (hereafter

named natural, Eflow and Eflow&Flush landscapes) in the hydrodynamic modeling of fish habitat in each study site.

## 2.5 Hydrodynamic modeling of fish habitat

The hydrodynamic modeling was performed using a calibrated version of the River2D model (Steffler et al., 2002). This is a finite element model widely used in fluvial modeling studies for the assessment of

habitat availability (Boavida et al., 2011; Jalón and Gortázar, 2007) that brings together a 2D hydrodynamic model and a habitat model to simulate the flow conditions of the river stretch and estimate its potential habitat value according to the fish habitat preferences. The strengths of this model are the fact of being public domain software and to be technically robust throughout a wide range of modeling circumstances. On the other hand, some limitations of this model are the non-incorporation of a morphodynamic module

or the ability of embodying fuzzy logic rules during the computation of species habitat availability.

The calibration procedure followed the methodology proposed by Boavida et al. (2013, 2015). Calibration was performed by iteratively adjusting the bed channel roughness to attain a good agreement of the simulated versus surveyed water surface elevations and velocity profiles in the surveyed cross-sections. Boundary conditions were set according to the water surface elevations measured at the upstream and

downstream cross-sections. Calibrated parameters are provided in supplementary material (Appendix A – Tables A1, A2, A3 and A4).

The hydrodynamic modeling comprised the Eflow discharge ranges in the study sites ($0 – 2$ m$^3$ s$^{-1}$ and $0 – 5.5$ m$^3$ s$^{-1}$ for OCBA and OCPR, respectively) and was accomplished for each riparian landscape scenario. The different riparian landscapes were represented in the hydrodynamic model by changing the channel

roughness according to the spatial extent of the riparian succession phases, i.e., the channel roughness inputted to the model are the riparian landscape maps converted into channel roughness maps. Roughness is a critical feature influencing the physical variables of flow hydraulics (Chow, 1959; Curran and Hession, 2013), whose distinct combinations typify diverse functional habitats, which are selected by fish according to its preference. The roughness classification of riparian vegetation succession phases was determined

based on roughness measurement literature on similar vegetation types (Chow, 1959; Wu and Mao, 2007) and expert judgment during model calibration.

After modeling the Eflow discharges in each of the riparian landscape scenarios of the two study sites, the hydraulic characteristics of each riparian landscape (roughness, flow depth and velocity) were compared using a t-test (confidence level of 99%) in R environment (R Development Core Team, 2011) in order to determine the existence of mean significant differences between riparian landscapes. Habitat simulation was achieved by the combination of the hydraulic modeling (flow depth and velocity) with preference curves information for the considered target species. The riverbed characteristics of substrate and cover were kept unchanged during the hydrodynamic modeling. Changing the substrate according to the modifications in succession phase disposal seemed to be an incorrect practice in this case because during data treatment, no significant differences were detected in riverbed substrate between succession phases. Cover modification was also disregarded because the CASiMiR-vegetation model only reproduces the riparian area, not the aquatic zone (note that this *aquatic zone* is a definition *sensu* CASiMiR-vegetation model, designating the area of the river channel that is permanently submerged throughout the hydrologic year and where riparian vegetation is unable to establish and develop. It corresponds to only a fraction of the wetted area by river flow during the discharges considered in the subsequent hydrodynamic modeling.) and therefore, this feature cannot be correctly modeled by the riparian vegetation model. Notwithstanding, the most important variables determining fish habitat availability influenced by riparian vegetation degradation were considered, namely, depth, velocity and substrate (Parasiewicz, 2007).

The Habitat Suitability Index (HSI) was determined for each species and life stage regarding the product of the velocity (Velocity Suitability Index – VSI), depth (Depth Suitability Index – DSI) and substrate (Substrate Suitability Index – SSI) variables, according with Eq. (1):

$$HSI = VSI \times DSI \times SSI \tag{1}$$

The product of the HSI by the influencing area (A) of the corresponding model $i^{th}$ node defines the Weighted Usable Area (WUA) of that node. The sum of the WUA's result in the total amount of habitat suitability for the study site, as described by Eq. (2):

$$WUA = \sum_{n=1}^{i} A_i \times HSI_i = f(Q) \tag{2}$$

Considering that the BACI approach (Before-After Control-Impact) is generally the best way of detecting impacts or beneficial outcomes in river systems (Downes et al., 2002) the resulting WUA's were then compared to the natural habitat in a census-based benchmark. The equality of proportions between habitat availabilities was tested using the $\chi^2$ test for proportions in R environment, while deviations were measured using the most commonly used measures of forecast accuracy, namely, Root Mean Square Deviation (RMSD), Mean Absolute Deviation (MAD) and Mean Absolute Percentage Deviation (MAPD). In all cases, smaller values of these measures indicate better performance in parameter estimation.

**2.6 Workflow of the modeling procedure**

The workflow of the modeling procedure is presented in Figure 3. Firstly, the calibrated version of the riparian vegetation model is used to produce the riparian landscape scenarios according to each of the considered flow regimes. In each modeling run, this model uses as inputs one of the specific flow regimes mentioned and models the effects of a decade of such flow regime in the local riparian vegetation. The

output of the model is an expected riparian vegetation landscape map (detailed by succession phases) resulting from the inputted flow regime. This map is converted into a channel roughness map by attributing to each riparian succession phase a specific effective roughness height based on the expert knowledge of the authors, on literature (e.g., Barnes, 1967; Chow, 1959; Fisher and Dawson, 2003) and on the calibration results of the models. The considered roughness values of each succession phase are provided as supplementary material (Appendix A – Tables A3 and A4). These roughness maps are one of the inputs of the River2D model.

Secondly, the hydrodynamic model River2D is used to determine the water depths and flow velocities at the microhabitat scale (already considering each of the roughness maps coming from the conversion of the CASiMiR-vegetation output vegetation maps) and to compute the weighted usable areas of the considered fish species using the previous calculated variables and the inputted information regarding the observed fish species habitat preferences for water depth and flow velocity. This is done similarly using every of the riparian landscape scenarios. For each scenario run, the outcome of this model is therefore the weighted usable area of each of the considered species and life stages for each of the discharges considered in the Eflow regime.

## 3 Results

### 3.1 Riparian vegetation modeling

Different riparian landscapes resulted from the riparian vegetation modeling according to the considered flow regimes in both case studies (Figure 4). Nonetheless, the modeled response of riparian vegetation to each flow regime is similar in the two study sites. The riparian landscape, driven by the natural flow regime, presents a river channel that is largely devegetated, where Initial (IP) and Pioneer (PP) phases together represent approximately 43% and 35% of the study site areas in OCBA and OCPR, respectively. In this riparian landscape, Early Succession Woodland phase (ES) can only settle in approximately 8% of OCBA and 1% of OCPR areas. The floodplain succession phases, namely, Established Forest phase (EF) and Mature Forest phase (MF), represent nearly 40 and 10% of the study area for OCBA and, close to 42% and 23% for OCPR, respectively.

In contrast, the riparian landscape created by the Eflow regime is where the riparian vegetation encroachment is more prominent. Herein, riparian vegetation settles in the channel and evolves toward mature phases due to the lack of the river flood disturbance. IP is now reduced to approximately 3% in OCBA and 6% in OCPR, while PP is inexistent in both cases. ES covers up to approximately 48% and 26% of the corresponding study areas, whereas EF and MF maintain about the same area in both case studies. The riparian landscape driven by the Eflow&Flush regime shows the capacity of this flow regime in hold back vegetation encroachment in both cases. In this riparian landscape scenario, IP and PP are maintained at approximately 30% of the study site area in both case studies, whereas ES is kept under 21% in OCBA and only 2% in OCPR. Once again, EF and MF preserve their areas in both case studies.

Summing up, the results of the riparian vegetation modeling show a riparian landscape degradation by vegetation encroachment in the Eflow landscape scenario when compared with the natural riparian landscape. Instead, the Eflow&Flush landscape scenario keeps approximately the same patch disposal and

succession phase's proportion as the natural landscape and therefore does not present evidence of riparian landscape degradation.

## 3.2 Hydrodynamic modeling

The changes undertaken by the riparian vegetation facing different flow regimes are able to modify the hydraulic characteristics of the river stretches (Figure 5). Channel effective roughness heights ($k_s$) change dramatically according to the considered riparian landscapes, increasing proportionally to the encroachment level of vegetation in the study sites. In both case studies, the $k_s$ values of the Eflow landscape are clearly distinct and higher compared to the other two riparian landscapes (Figure 5). The $k_s$ values in the Eflow&Flush landscape were found to be between the values of Eflow and natural landscapes in the case of OCBA, and were very similar with the natural landscape in the case of OCPR (Figure 5). Notwithstanding, in both case studies, the $k_s$ mean values are statistical significantly different between all three riparian landscapes (test results in supplementary material; Appendix C – Table C1). The mean $k_s$ of the Eflow, Eflow&Flush and natural landscapes are 0.999, 0.709 and 0.462 m, respectively, in OCBA, and 1.034, 0.742 and 0.7178 m, respectively, in OCPR.

Changes also occur in flow depth and flow velocity for the considered discharge range of the proposed environmental flows (Figure 5). Although not so noticeable due to the great amount of data, differences are statistically significant. In OCBA, the Eflow landscape creates a circumstance with statistically significant higher depths (mean depth is 0.402 m) and lower flow velocities (mean flow velocity is 0.128 m s$^{-1}$) than the natural and Eflow&Flush landscapes. The t-tests on water depths (H0: true difference in means is equal to 0) revealed highly significant p-values ($<0.001$), respectively, for the comparisons between Eflow and natural flow regimes, and Eflow and Eflow&Flush flow regimes. The t-tests on flow velocities also derived a highly significant p-value ($<0.001$) in both the comparisons of natural versus Eflow regimes and Eflow versus Eflow&Flush flow regimes (test results in supplementary material; Appendix C – Tables C2 and C3). In contrast, depth and flow velocity are not significantly distinguishable between the natural and Eflow&Flush landscapes, where mean depth and flow velocity are 0.397 m and 0.136 m s$^{-1}$, respectively, in the former, and 0.399 m and 0.135 m s$^{-1}$ respectively, in the latter.

For the OCPR study site, flow depths are not significantly different (t-tests obtained p-values of 0.122 for natural versus Eflow regimes and 0.098 for Eflow versus Eflow&Flush flow regimes). Mean values of flow depth for Eflow, Eflow&Flush and natural landscapes are 0.420, 0.417, 0.418, respectively. Nonetheless flow velocities are different with statistical significance as the p-values of the t-tests for natural versus Eflow and for Eflow versus Eflow&Flush were highly significant ($<0.001$). The Eflow landscape creates statistical significantly lower flow velocities (0.271 m s$^{-1}$) when compared to the statistical significantly indistinct Eflow&Flush (0.277 m s$^{-1}$) and natural (0.278 m s$^{-1}$) landscapes (test results in supplementary material; Appendix C – Tables C2 and C3).

Furthermore, when comparing water depths and flow velocities point by point, one can find differences between scenarios up to 10 cm in water depth and more than 40 cm s$^{-1}$ in flow velocity. Accordingly, there are locations where the considered hydraulic parameters change considerably, shifting the habitat preference of fishes in one or two classes of the corresponding habitat preference curves.

In general, the Eflow landscapes present an increased channel roughness interfering with river flow and creating increased water depths and slower flow velocities when compared with the natural landscape. On the contrary, despite the increased channel roughness of the Eflow&Flush landscape, the water depths and flow velocities are very similar to the ones in the natural landscape. These results demonstrate that an environmental flow addressing exclusively fish requirements is not capable of preserving the habitat availability of the aquatic species for which was proposed in the long-term.

### 3.3 Analysis of the aquatic habitat suitability for fish species

During a hydrological year, each riparian landscape provides different WUAs for the target fish species, with the same environmental flow regime addressing fish species (Figure 6). Differences from the natural habitat suitability are greater in the Eflow landscape for both case studies. In OCBA, major differences in the WUA can be found almost all year round for the barbel juveniles, throughout autumn and winter months for the nase juveniles and during spring months for the calandino. Compared to the natural landscape, the WUA modifications instilled by the Eflow landscape are on average approximately 12%, and are higher than 17% in a quarter of the cases reaching 80% in an extreme situation. Particularly, the Eflow landscape provides less habitat suitability during autumn and winter months for the barbel and nase juveniles, c. 17% and 14%, respectively. Likewise, in this riparian landscape, the habitat suitability during spring months increases approximately 23% for the barbel juveniles and approximately 20 and 27% for the calandino juveniles and adults, respectively. On the other hand, throughout the year, the Eflow&Flush landscape provides a WUA very similar to the natural landscape. The habitat changes created by the Eflow&Flush landscape are on average approximately 2% and never reach 8% for all species and life stages.

As for OCPR, major differences in WUA are seen almost all year round for calandino and nase, and exist particularly in spring months for barbel. WUA modifications due to the Eflow landscape are on average near 29%, being a quarter more than 50% and reaching up to more than 100% different in the most extreme case. The Eflow landscape consistently provides less habitat suitability during autumn and winter months for the nase juveniles and adults, c. 50% and 38%, respectively, while the habitat suitability increases in approximately 46% of calandino. Moreover, the Eflow landscape provides an increased WUA during spring months in approximately 18% of the barbel adults and 71% of the calandino adults, while it decreases the habitat on average for approximately 7% of the remaining species and life stages. Also in this case study, the Eflow&Flush landscape provides a WUA very similar to the natural landscape throughout the year. The habitat changes created by the Eflow&Flush landscape are on average near 3% and always less than 17% for all species and life stages. Accordingly, in both case studies, the WUA differences evidenced in the Eflow landscape revealed to be significant in several months by the $\chi^2$ test whereas this were never the case for the Eflow&Flush landscapes (test results provided in supplementary material; Appendix C – Tables C4, C5, C6 and C7).

The riparian-induced modifications on the WUAs are also confirmed by all the employed deviation measures (Table 2). According to RMSD, MAD and MAPD, the habitat provided by the Eflow landscape is always farther apart from the natural habitat for all species and life stages. In OCBA, the larger deviations occur for the barbel juveniles and nase adults, whereas in OCPR, the calandino adults and the barbel juveniles are the ones enduring greater habitat deviations from the natural circumstance. All together, these

results reveal that the overlook of riparian requirements into environmental flows can derail the goals of environmental flows addressing only aquatic species by an extent of approximately an average of 12 to 29% of the fish WUA's in the considered study sites as a result of the riparian landscape degradation. On the other hand, results reveal that environmental flows regarding riparian requirements are able to maintain the habitat availability of fish species as the WUA's in the study sites never change on average more the 3% in a decade.

## 4 Discussion

This study evaluated the benefits of incorporating riparian requirements into environmental flows by estimating the expected repercussions of riparian changes driven by regulated flow regimes on the fish long-term habitat suitability. To this end, the riparian vegetation was modeled for 10-year periods according to three different flow regimes and results were inputted as the habitat basis for the hydrodynamic modeling and subsequent assessment of the fish habitat suitability in those riparian landscapes. Such ecological modeling approach, where a joint analysis is performed while embracing a suitable time response for the ecosystems involved, enables a realistic biological-response modeling and substantiates the long-term research that is required in environmental flow science (Arthington, 2015; Petts, 2009). Furthermore, this approach allows one to foresee and assess the outcome of recommended flow regimes, which is an essential topic but has been poorly considered in environmental flow science (Davies et al., 2013; Gippel, 2001). This research provides an insight of the expected long-term effects of environmental flows in river ecosystems, therefore unveiling the potential remarkable role of riparian vegetation on the support of environmental flows efficiency, which can transform the actual paradigm in environmental flow science.

During modeling, geomorphology was considered immutable and sediment transport originated by the environmental flow regimes was disregarded. River morphodynamics and its interactions with riparian vegetation constitute an important river process in many rivers, particularly in fine sediment rivers (e.g., Corenblit et al., 2009; Corenblit et al., 2011; Gurnell et al., 2012; Gurnell, 2014). However, the research on the temporal scales of geomorphic and ecological processes is still scarce in coarse-bed rivers (Corenblit et al., 2011), and simultaneously more complex and uncertain (Yasi et al., 2013). The error predictions from best hydraulic predictors in this type of rivers can range between 50 to 200% (Van Rijn, 1993; Yasi et al., 2013). Disregarding such processes in these study sites was carefully considered. Given the above and the arguments mentioned in the methods section, we are confident that this option in this case will not bring tangible shortcomings to this research. Furthermore, the possible riverbed degradation effects due to the releasing of sediment-starving floods by the dam were not tested because according to our expert knowledge this will not pose a problem in this case. Such floods with similar recurrence intervals were already tested by Rivaes et al. (2015) in two river stretches of much smaller grain size (pebbles and sand) and results showed in both cases that such flood discharges were not relevant for riverbed degradation. The influence of fish species on geomorphology and riparian vegetation by ecosystem engineering, as it was mentioned in the introduction, was not considered also during this study as it seemed fairly unrealistic in these case studies due to the general dimension of riverbed particles.

The results of the vegetation modeling illustrate how the natural flow regime generates morphodynamic disturbances, without which the riparian vegetation is able to settle and age in the river channel. This is an

important outcome that is essential to remember when providing environmental flow instructions. Subsequently, microhabitat analysis demonstrated that changes in the riparian landscape induce modifications in the hydraulic characteristics of the river stretches. The differences in mean values of these parameters are subtle between riparian landscapes but are statistically significant. Furthermore, a detailed analysis using a pairwise comparison of flow depths and velocities between scenarios show that modifications can reach 10 cm in water depth and more than 40 cm s$^{-1}$ in flow velocity in some places. The hydrodynamic modeling results show that the water flowing near the margins is more affected than the water flowing in deeper areas of the river channel. One reason for these results is certainly because this study is about the effects of riparian vegetation encroachment on the physical habitat due to the colonization of the river margins by woody riparian vegetation.

Accordingly, there are locations where the considered hydraulic parameters change considerably, shifting the habitat preference of fishes in one or two classes of the corresponding habitat preference curves. Such change can shift the habitat preference of fishes in one or two classes of the corresponding habitat preference curves. These changes are particularly important considering that an alteration of one class regarding these parameters is sufficient to change fish preferences from near null to maximum and vice-versa in many cases, as it can be seen in the preference curves provided in the supplementary material (Appendix B – Figures B10, B11 and B12).

The hydrodynamic modeling also indicated changes directly affecting the habitat suitability of the existing fish species according to the riparian landscape. Through time, the riparian landscape shaped by the Eflow regime diverged in habitat suitability from the natural and Eflow&Flush landscapes, and there were cases where the habitat suitability was modified by more than double. The relationship between fish assemblages and habitat has long been acknowledged (e.g., Clark et al., 2008; Matthews, 1998; Pusey et al., 1993) and can have a significant impact on the ecological status and function of the existing fish communities (Freeman et al., 2001; Jones et al., 1996; Randall and Minns, 2000). Effectively, habitat loss is the major threat concerning fish population dynamics and biodiversity (Bunn and Arthington, 2002), thereby promoting population changes with a proportional response to the enforced habitat change (Cowley, 2008). This is particularly true for the fish species considered in this study (Cabral et al., 2006). The habitat decrease for barbel and nase during autumn and winter months jeopardizes those species survival by refuge loss, which is particularly important in flashy rivers (Hershkovitz and Gasith, 2013), such as the Ocreza river and Mediterranean rivers in general. On the other hand, the habitat change during spring months undermines the spawning activity and consequently the sustainability of future population stocks (Lobón-Cerviá and Fernandez-Delgado, 1984). The habitat increase of calandino during this period can be ecologically tricky due to the habitat plasticity of this species (Doadrio, 2011; Gomes-Ferreira et al., 2005), as well as its characteristic adoption for an r-selection strategy as an evolutionary response to frequently disturbed environments (Bernardo et al., 2003). Above all, one should not ignore that the relationships between fish assemblages and habitat are extremely complex (e.g., Diana et al., 2006; Hubert and Rahel, 1989; Santos et al., 2011), being a consequence of the actual natural conditions (Poff and Allan, 1995; Poff et al., 1997) that when disrupted, may allow the expansion of more generalist and opportunistic fauna (Poff and Ward, 1989).

Our results indicate that environmental flows taking into account riparian vegetation requirements are able to preserve the naturalness of the riparian landscape and consequently, the maintenance of the fish habitat suitability. Accordingly, the implementation of such measure in place of using environmental flows addressing only fish requirements can provide significant positive ecological effects in downstream reaches (Lorenz et al., 2013; Pusey and Arthington, 2003) and additional ecosystem services like stream bank stability, flood risk reduction or wildlife habitat (Berges, 2009; Blackwell and Maltby, 2006) while imposing minor revenue losses to dam managers (Rivaes et al., 2015).

The implementation of such environmental flows could provide an additional way to attain the "good ecological status" required by the Water Framework Directive (WFD). In addition, taking up a procedure such as this one can act both as 'win-win' and 'no-regret' adaptation measures during the second phase of the WFD, because it potentiates the improvement of other ecological indicators and mitigates the impacts of flow regulation, while being robust enough to account for different scenarios of climate change (EEA, 2005).

Water science still lacks strong links between flow restoration and its ecological benefits (Miller et al., 2012), particularly regarding long-term monitoring of environmental flow performance (King et al., 2015 and citations herein). Nevertheless, the outcomes of this study are a product of long-term simulations by models that were calibrated and validated for the corresponding watershed with local data in natural river flow conditions. This standard procedure in modeling strengthens confidence in our predictions as the models proved to correctly replicate the response of the riparian and fish communities when paralleled with simultaneous observational data. In addition, model uncertainty due to estimation uncertainty in input parameters was previously assessed by means of sensitivity analyses on both models. In either case the models showed to be quite robust to the uncertainty of estimated parameter inputs (see Rivaes et al., 2013 and Boavida et al., 2013) which reveal a relatively small uncertainty in the models outputs and provides additional confidence on the results.

In conclusion, we predict a change in fish habitat suitability according to the long-term structural adjustments that riparian landscapes endure following river regulation. These changes can be attributed to the effects that altered riparian landscapes have on the hydraulic characteristics of the river stretches. In our view, environmental flow regimes considering only the aquatic biota are expected to become obsolete in few years due to the alteration of the habitat premises in which they were based. This situation points to the unsustainability of these environmental flows in the long-term, failing to achieve the desired effects on aquatic communities to which those were proposed in the first place. An environmental flow regime that simultaneously considers riparian vegetation requirements contributes to the preservation of the hydraulic characteristics of the river channel at the natural riverine habitat standards, therefore maintaining the habitat assumptions that support the environmental flow regimes regarding aquatic communities. Consequently, accounting for riparian vegetation requirements poses as an essential measure to assure the effectiveness of environmental flow regimes in the long-term perspective of the fluvial ecosystem.

**Data availability**

Riverbed topography, hydraulic measurements, riparian vegetation and fish sampling were collected by the authors and are available at http://doi.org/10.5281/zenodo.839531. Both River2D and CASiMiR-vegetation

models are freeware available at http://www.river2d.ualberta.ca/download.htm and http://www.casimir-software.de/ENG/download_eng.html, respectively.

**Acknowledgments**

This research was financially supported by Fundação para a Ciência e a Tecnologia (FCT) under the project UID/AGR/00239/2013. Rui Rivaes benefited from a PhD grant sponsored by FCT (SFRH/BD/52515/2014). Isabel Boavida was supported by a post-doctoral grant (SFRH/BPD/90832/2012) also sponsored by FCT. José Maria Santos was supported by a postdoctoral grant from the MARS project (http://www.mars-project.eu). The Portuguese Institute for Nature Conservation and Forests (ICNF) provided the necessary fishing and handling permits.

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

**Table 1. Maximum annual discharges (m$^3$ s$^{-1}$) considered in the CASiMiR-vegetation model for each study site.**

| Year | OCBA | | | OCPR | | |
|------|---------|-------|-------------|---------|-------|-------------|
|      | natural | Eflow | Eflow&Flush | natural | Eflow | Eflow&Flush |
| 1    | 671     | 0.99  | 0.99        | 951     | 5.51  | 5.51        |
| 2    | 203     | 0.99  | 167         | 287     | 5.51  | 237         |
| 3    | 327     | 0.99  | 0.99        | 464     | 5.51  | 5.51        |
| 4    | 217     | 0.99  | 167         | 308     | 5.51  | 237         |
| 5    | 316     | 0.99  | 0.99        | 449     | 5.51  | 5.51        |
| 6    | 371     | 0.99  | 167         | 526     | 5.51  | 237         |
| 7    | 702     | 0.99  | 0.99        | 995     | 5.51  | 5.51        |
| 8    | 202     | 0.99  | 167         | 286     | 5.51  | 237         |
| 9    | 195     | 0.99  | 0.99        | 276     | 5.51  | 5.51        |
| 10   | 440     | 0.99  | 371         | 624     | 5.51  | 527         |

**Table 2. Deviation analysis of the weighted usable areas for the considered regulated flow regimes benchmarked by the natural flow regime (RMSD – Root Mean Square Deviation, MAD – Mean Absolute Deviation, MAPD – Mean Absolute Percentage Deviation). Values stand for the habitat availability deviation, in area and percentage, of the environmental flow regimes compared to the natural habitat availability of each species and life stage.**

| | OCBA study site | | | | | | OCPR study site | | | | | |
| --- | --- | --- | --- | --- | --- | --- | --- | --- | --- | --- | --- | --- |
| | Eflow | | | Eflow&Flush | | | Eflow | | | Eflow&Flush | | |
| | RMSD | MAD | MAPD | RMSD | MAD | MAPD | RMSD | MAD | MAPD | RMSD | MAD | MAPD |
| | ($m^2$) | ($m^2$) | (%) | ($m^2$) | ($m^2$) | (%) | ($m^2$) | ($m^2$) | (%) | ($m^2$) | ($m^2$) | (%) |
| *Luciobarbus bocagei* (juv.) | 86.00 | 72.10 | 15.40 | 12.17 | 7.24 | 2.52 | 26.23 | 17.37 | 35.55 | 2.51 | 1.50 | 0.63 |
| *Luciobarbus bocagei* (adult) | 29.46 | 20.55 | 5.83 | 2.87 | 2.12 | 1.55 | 12.94 | 7.73 | 23.15 | 3.44 | 1.79 | 3.01 |
| *Pseudochondrostoma polypepis* (juv.) | 128.21 | 86.14 | 11.58 | 9.42 | 5.72 | 2.26 | 45.42 | 32.71 | 34.43 | 1.55 | 0.92 | 2.51 |
| *Pseudochondrostoma polypepis* (adult) | 7.32 | 5.85 | 18.70 | 2.17 | 1.37 | 2.10 | 9.00 | 7.00 | 10.34 | 0.51 | 0.35 | 2.42 |
| *Squalius alburnoides* (juv.) | 44.05 | 28.16 | 8.46 | 6.20 | 4.06 | 2.10 | 33.10 | 27.78 | 28.37 | 2.44 | 1.35 | 2.18 |
| *Squalius alburnoides* (adult) | 92.41 | 52.47 | 10.23 | 7.49 | 5.31 | 2.37 | 61.76 | 47.83 | 40.54 | 0.96 | 0.63 | 2.90 |

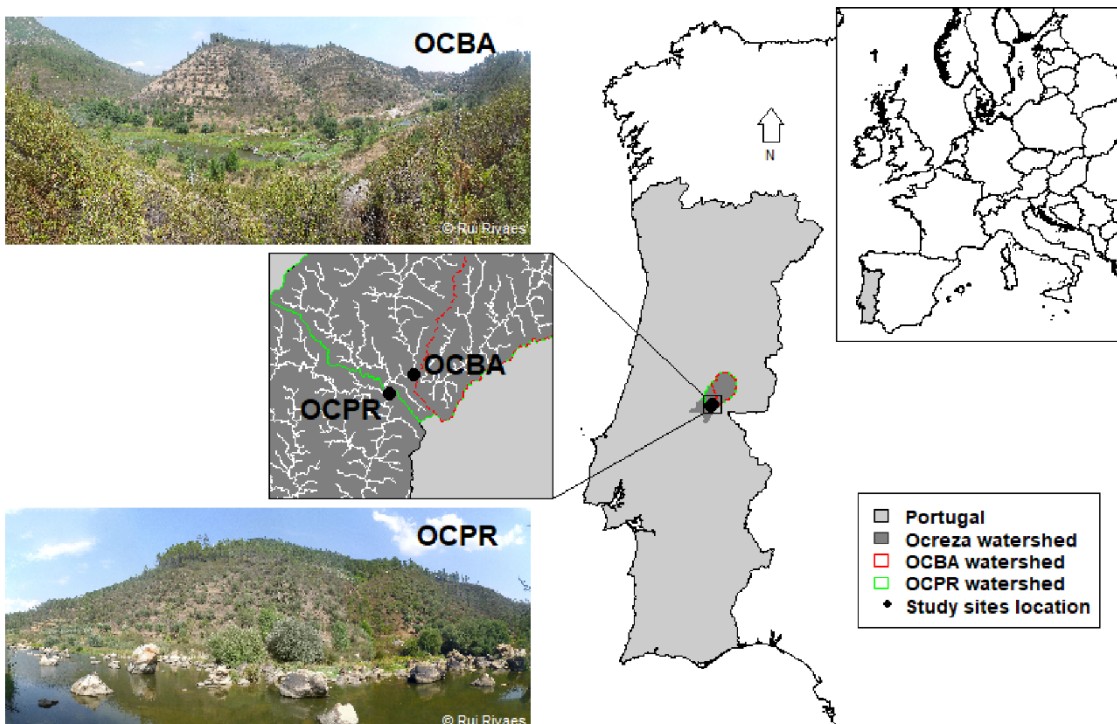

**Figure 1. Location and characterization of the study sites OCBA and OCPR.**

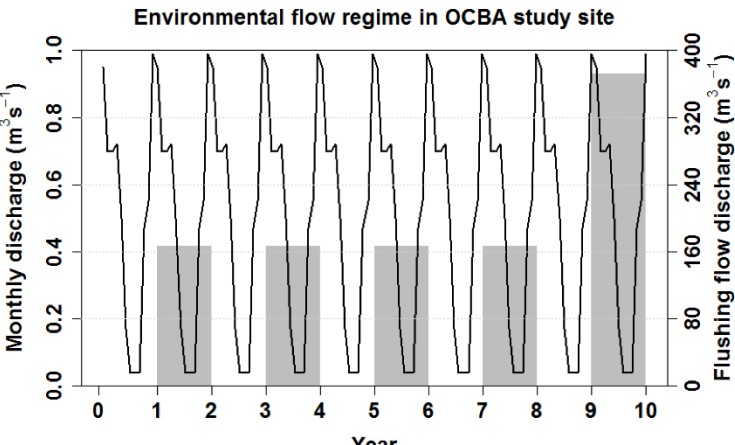

**Figure 2. Environmental flow regime addressing fish (black line, left axis) and riparian (grey bars, right axis) requirements considered for the habitat modeling in OCBA study site. Fish requirements are addressed by a constant monthly discharge and riparian requirements by a flushing flow in the years in which are planned (duration of the flushing flow is similar to a natural flood with equal recurrence interval). The hydrograph for the Eflow&Flush flow regime is similar in the OCPR study site.**

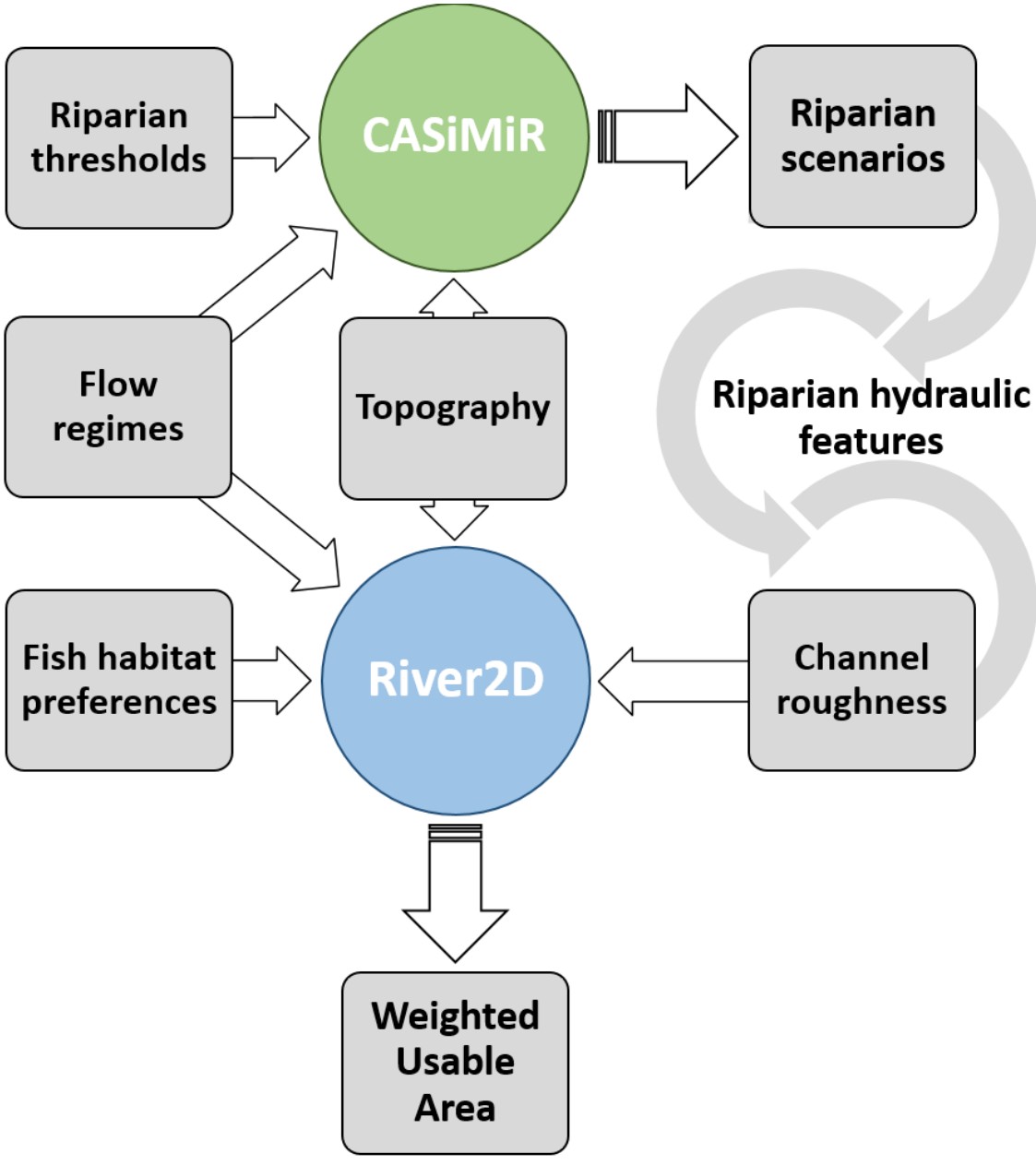

**Figure 3. Methodological scheme representing the workflow of the modeling procedure. White arrows stand for direct inputs, striped white arrows for model outputs and grey arrows for variable conversion processes.**

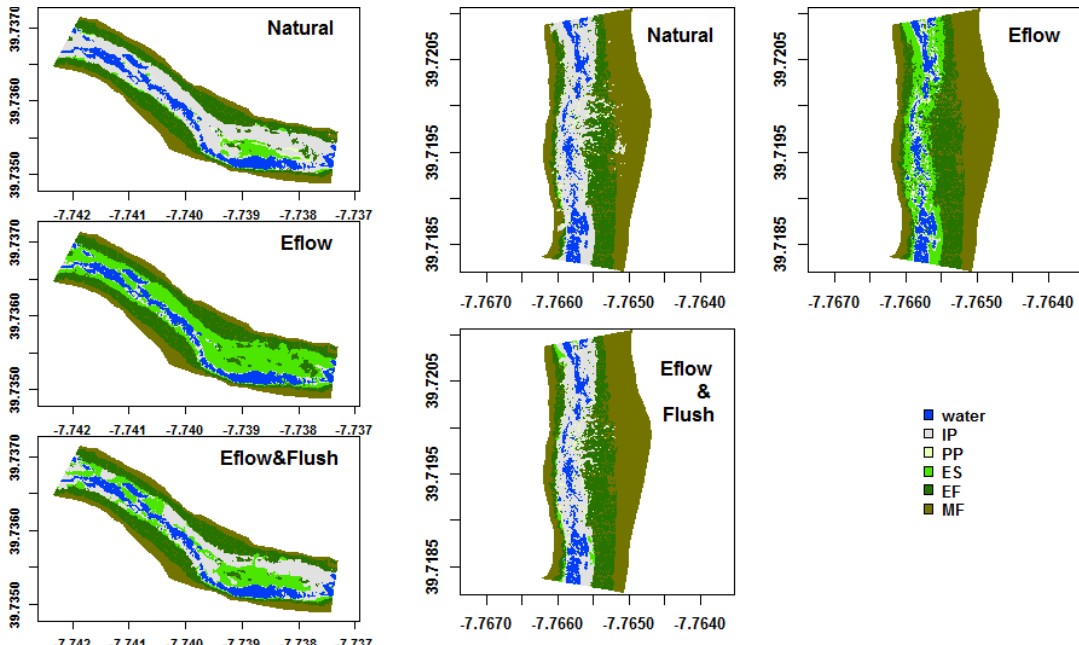

**Figure 4. Expected patch mosaic of the riparian vegetation habitats shaped by the natural, Eflow and Eflow&Flush flow regimes (detailed by succession phase, namely, initial phase – IP, pioneer phase – PP, early succession woodland phase – ES, established forest phase – EF and mature forest phase – MF) in the OCBA study site (on the left) and in the OCPR study site (on the right).**

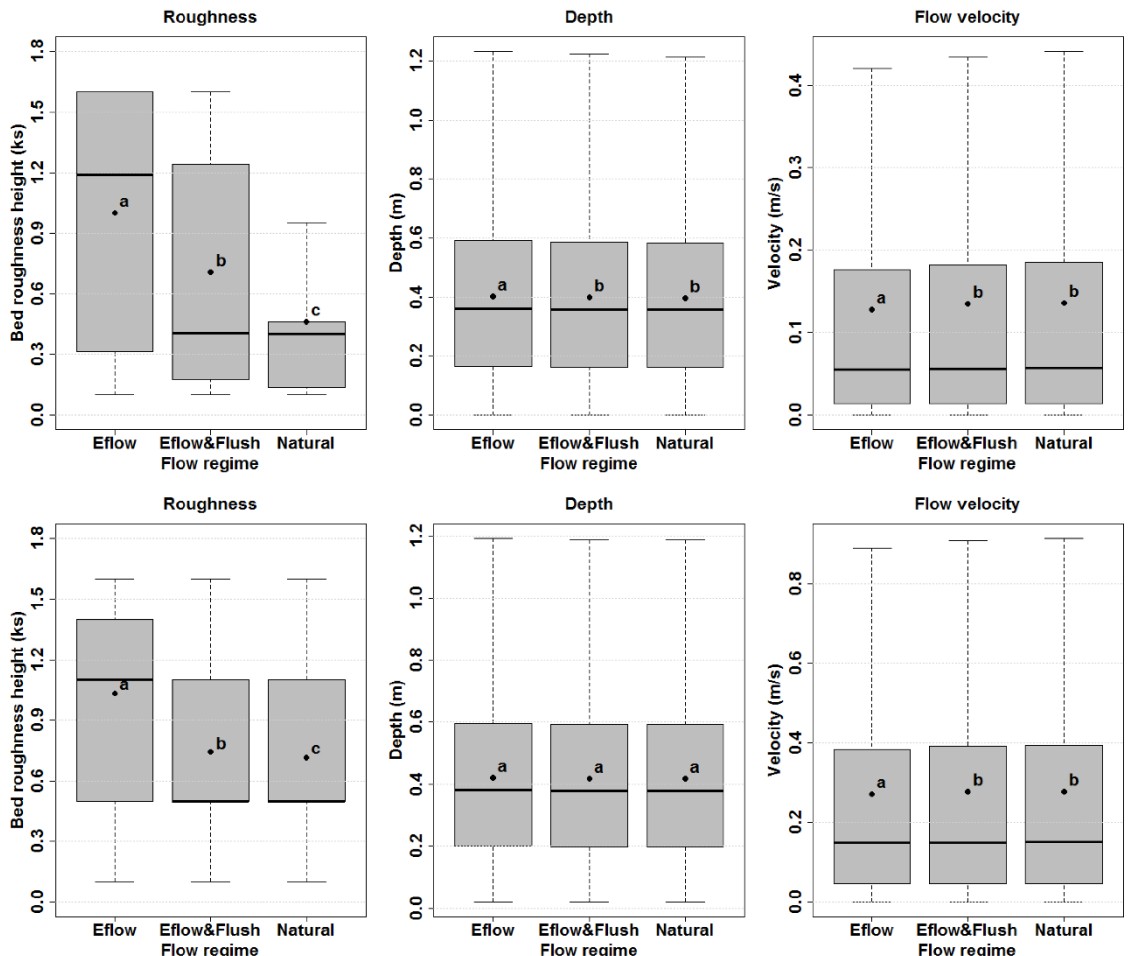

**Figure 5. Hydraulic characterization of OCBA (top) and OCPR (bottom) according to the different expected riparian vegetation habitats driven by the Eflow, Eflow&Flush and natural flow regimes (data obtained from 2D hydrodynamic modeling). Different letters stand for statistical significant differences between groups (t-test). Boxplots portray non-outlier value range, thick black lines the median value and black dots the mean values.**

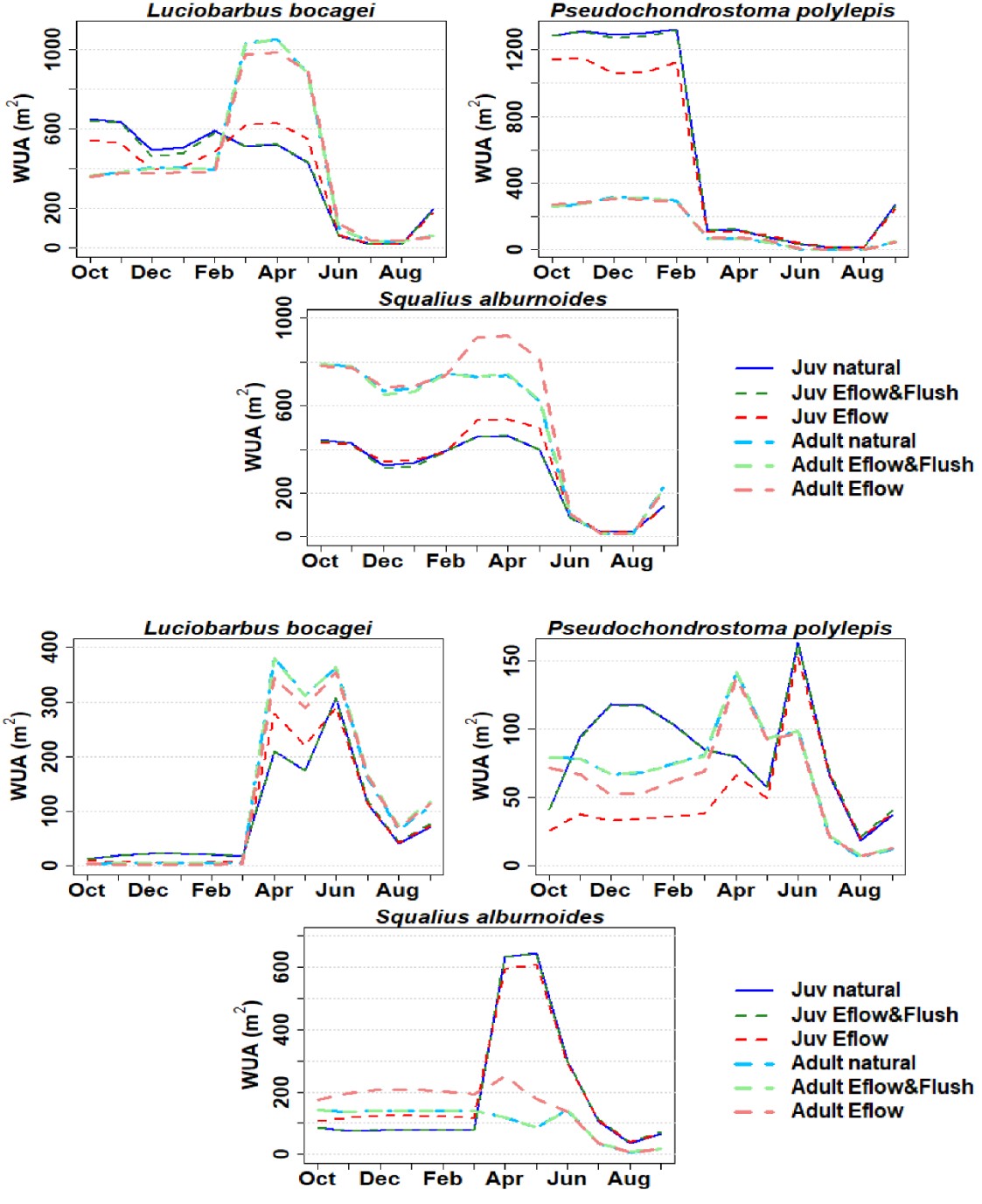

**Figure 6. Fish weighted usable areas provided by the fish-addressed environmental flow regime (Eflow) flowing through the different riparian landscape scenarios originated by a decade of three different flow regimes (natural, Eflow&Flush and Eflow) at the OCBA (top three graphics) and OCPR (bottom three graphics) study sites.**