# Peer review of "Importance of considering riparian vegetation requirements for the long-term efficiency of environmental flows"

_Hydrology and Earth System Sciences, 2017_

## Referee Comment (RC1) · Anonymous Referee #1 · 10 May 2017

General comment: This manuscript presents a research modelling the effects of environmental flow regimes implementation, with the novelty of considering ecological requirements of riparian vegetation as an alternative to traditional environmental flows generally based on the requirements of a single biological group, mostly fish. The analysis is applied to two reaches (<500 m) located in the Ocreza River that, although very close between them, their catchment areas are very different as well as its general valley typology. Authors employ vegetation and hydrodynamic modelling techniques together with valuable field data of riparian vegetation and fish communities, under three different flow regimes: natural, environmental flows only considering fish requirements and a third environmental flow regime incorporating flushing flows very important for

riparian vegetation persistence. Results show that environmental flows disregarding riparian vegetation requirements promoted vegetation encroachment, while the consideration of flushing flows were able to maintain riparian vegetation near natural standards in addition to maintain fish habitat availability similar to natural habitat (change less than 16.17 %).

This work is very stimulating and potentially of great interest as it encourages the consideration of other biological groups in environmental flows design. However, I think it should be improved in some aspects:

My main concerns are the following: 1. The methodological section should include some clarifications about the structure of modelling applications. It is not clear what authors obtained from one model and use as input for the other model. I think that a methodological scheme specifying steps in boxes would greatly improve the understanding which is crucial for research reproducibility. If it is no possible because of pages limit, some indications should be included in the text. My main question is: after obtain three different habitats configuration from the three flow regime modellings, how did authors applied the hydrodynamic modelling to calculate WUAs? If CASiMiR-vegetation model only reproduces the riparian area, not the aquatic zone (as authors said in page 6 lines 37-38), and also uses a fixed topography, why authors expect that conditions for fish change in the aquatic area? Explanations about how the models work, how are they connected and limitations of both are needed. For example in page 9 Lines 25-28, authors said "Consequently, of the latter, the microhabitat analysis demonstrated that changes in the riparian habitat induce modifications in the hydraulic characteristics of the river stretches". If the aquatic zone is not modify by CASiMiR and fish live in aquatic zone (punctually in other zones when zones are flooded), does it matter if the hydraulic characteristics of the riparian corridor covered by woody vegetation are modified? 2. About flow regime definition (section 2.3) authors mentioned that "the considered environmental flow regimes were adapted from the environmental flow regime proposal for the future Alvito dam (Ferreira et al, 2014)", but in the paragraph

after authors mentioned that "Eflow was determined according to the Instream Flow Incremental Methodology (Bovee, 1982)". Then, how was really? They are adapted or they were created for this paper? Or they were created in Ferreira et al., 2014 according to Bovee (1982) methodology and used here? It is no clear. 3. Regarding environmental flows considering riparian vegetation, I know the paper from Rivaes et al. (2015) that propose flushing flows to maintain the ecological succession equilibrium of riparian vegetation. Have authors think about the consequences for bed channel that would exist if a very small discharge is maintained along the time (in your case 0.99 m3s-1), and after two years dam release a high discharge of "clean water" without any sediment? Those floods are probably going to produce incision in the main channel. Then vegetation encroachment may be avoid, but with catastrophic consequences for the main channel in my opinion. Discussion about this type of limitations will enrich the paper and contextualize the results. 4. Regarding vegetation modelling, CASiMiR model lacks of a crucial process such as the morphological evolution of the river as it uses a fixed topography. The interaction between river morphodynamics and riparian vegetation has been widely studied, with bi-directional influences. Riparian vegetation affects channel morphology and flow dynamics affect riparian vegetation. Then, given that in the ten years of modelling some floods occur, the sentence (page 6, lines 9-11) "Such modeling period was considered to be long enough to avoid the influence of the initial vegetation conditions, while river morphological changes still do not assume importance in vegetation development (Politti et al., 2014)" is not truly appropriate, nor the sentence neither the reference. The reference is not appropriate because Politti et al., (2014) applied also CASiMiR model although in a climate change context. Then, in the case that they conclude this (I think that they don't conclude this), I disagree because using a fixed topography for a vegetation-modelling is incompatible with that conclusion unless they compare with a variable topography, which is not possible in CASiMiR-vegetation model. The sentence is not appropriate because there is a wide list of literature that that say the contrary. For example: Corenblit, D., Baas, A.C., Bornette, G., Darrozes, J., Delmotte, S., Francis, R.A., Gurnell, A., Julien, F., Naiman,

R.J., Steiger, J., 2011. Feedbacks between geomorphology and biota controlling Earth surface processes and landforms: A review of foundation concepts and current understandings. Earth-Science Reviews 106 (3-4), 307–331. Corenblit, D., Steiger, J., Gurnell, A., Naiman, R.J., 2009. Plants intertwine fluvial landform dynamics with ecological succession and natural selection: a niche construction perspective for riparian systems. Global Ecology and Biogeography 18 (4), 507–520. Gurnell, A., 2014. Plants as river system engineers. Earth Surface Processes and Landforms 39(1), 4-25 Gurnell, A., Bertoldi, W., Corenblit, D., 2012. Changing river channels: The roles of hydrological processes, plants and pioneer fluvial landforms in humid temperate, mixed load, gravel bed rivers. Earth-Science Reviews 111(1), 129-141.

5. About results presentation, now this section is a bit confuse and I think it will benefit from the emphasis of main results, for example, answering explicitly to the questions that authors propose at the end of introduction section.

Specific comments:

Title: As your study encompass a decade, talk about "the long-term" is not very appropriate. Paper from Frissel et al., (1986) relate the "reach system" (that could be equivalent to reaches in the paper) to a time scale of 10 to 100 years. Then, authors choose the minimum threshold, not really the long-term. Authors should avoid using that expression for a period of 10 years with riparian vegetation context.

Introduction: Introduction section provides an appropriate "stat-of-the-art" about the main topic. As authors have been able to formulate the objectives as questions, they should take the advantage and give results to clearly answer those questions.

Methods: Study site: Page 3 line 9: Authors use a very general reference to talk about the flow regime of a typical Mediterranean river. Are there discharge data of the river? Because the sentence should describe the real flow regime instead of use a general reference. As in "Flow regime definition" authors use a "natural regime" and also flushing floods, information about return periods also should appear in the

description of the study area. Figure 1. Figure 1 is not very informative. The zoom of drainage network could include catchment delimitation of each reach. Caption figure should include photos authorship.

Data collection Please, give a brief description about field procedures like for example if electrofishing was used, although all details can be seen in Boavida et al., (2011).

Riparian vegetation modelling Page 5 lines 36-39: "the hydrological regime is inputted into the model in terms of maximum annual discharges as these discharges are considered as the annual threshold for riparian morphodynamic disturbance that determine the succession or retrogression of vegetation." In this case with extreme low flows proposed in Eflows, dessication could be also dcrucial for vegetation retrogression. Not only minimum discharges (quantity) but also duration. Have the model consider that? If not, please explain explicitly. Page 6 Line 8: Authors have included many supplementary material which is very appreciate. But, please specify the supplementary material in each case along the entire manuscript, i.e. the number of table or figure because otherwise is confuse. Page 6 Lines 12-13: "The resulting riparian. . .hereafter named natural, Eflow and Eflow&Flush habitats". The word "habitats" is ambiguous because in each of this flow regime there are "habitats". . .I suggest using "scenarios" or something similar instead of "habitats". ". . .hereafter named natural, Eflow and Eflow&Flush scenarios habitats". Please, check and be congruent along the manuscript.

Table S5 in supplementary material contains some of the vegetation model parameters. The Resistance to shear stress (N m-2) that authors used differs greatly from parameters used by Politti et al., (2014). While the current paper use 30, 30, 50, 300 and 300 N m-2 for IP, PP, ES, EF and MF respectively, Politti et al., (2014) used 1, 3, 40, 25, 60 and 400 as critical values of shear stress for "Initial phase", "Pioneer phase", "Herb phase", "Pioneer shrub", "Shrub phase" and "Early successional woodland" . The classes and not totally equivalent, but why are they so much different?

Hydrodynamic modeling Page 6 Lines 29-31. "The hydraulic characteristics of each

habitat (roughness, flow depth and velocity) were compared using a t-test (confidence level of 99%) in R environment (R Development Core Team, 2011) in order to determine the existence of mean significant differences between habitats." I understand that authors are looking for significant differences between scenarios (natural, Eflow and Eflow&Flush). Then, did authors mix hydraulic values (water depth, water velocity and roughness) from different zones (IP, PP...etc.) for all riparian area in each scenario? For what purpose? Are fish going to use EF or MP zone? Why not looking for differences between scenarios considering the different zones (IP, PP, etc.)? Knowing those differences would be more interesting that the main value for all riparian area.

Results: In general, this section could be better structured with some sub-sections. Also, main results should be explicitly written, for example as brief conclusions for each part. I mean, along the text there are many comparisons in %, but a sentence summarizing what they mean in general terms when it is possible (the consequences of applying one flow regime or other) would greatly improve the understanding for readers, it is a suggestion. Page 7 Line 19: Here authors use "habitat" in a different context. That is why I recommend using "scenario". Page 7 Line 36: "The changes undertaken by the riparian vegetation facing different flow regimes are able to modify the hydraulic characteristics of the river stretches (Figure 4)". Do these values (Figure 4) refer to the entire riparian area in each scenario? As in my previous comment about "hydro-dynamic modelling", I suggest that results be presente for each zone because not all zones affect fish habitat. Page 8 Lines 3-6: Comments about ks, it is not clear which comment refers to figure 4 and which comment refers to supplementary material. Figure 4 and Table S8, S9, S10 in supplementary material contain the same information? I mean, figure 4 graphically and tables with tests? It is not clear what authors try to differentiate with tests. Please, include some clarifications in the text. Figure 5: It is not indicated to which reach refer each set of graphics. I suppose that upper graphics are from OCBA and the lowers from OCPR. Please, indicate it. Also regard to this figures, the colors and line types used are not truly appropriate. Authors should use line types that are distinguishable when they overlap. And the thickest line should be finer. Page

8 Line 34: "The Eflow habitat consistently provides less habitat suitability during autumn and winter months for the barbel and nase, c. 50% and 38%, respectively, while the habitat suitability increases in approximately 46% of calandino." May be is because graphics are not adequately labelled, but the "less suitability for the barbell during autumn and winter" is not really lower. In the graphic it seems the same. Please, check that text describes correctly the graphics.

Discussion: As I mentioned before, the modelling techniques that authors used have some limitations that should be comment in the discussion section. Page 9 Lines 27: Authors use "habitats" and in line 28 use "scenarios". Please, authors should homogenize the terms used. Page 10 line 19: "Accordingly, the implementation of such measure can provide significant positive ecological effects in downstream reaches (Lorenz et al., 2013; Pusey and Arthington, 2003) and results in additional ecosystem services (Berges, 2009; Blackwell and Maltby, 2006) while imposing minor revenue losses to dam managers (Rivaes et al., 2015)." Significant positive ecological effect… compared to what? To the natural regime??? To the Eflow regime? What type of ecosystem services? Could you specify? Please, rewrite this paragraph because it is not clear.

---

## Referee Comment (RC2) · M. Ertsen (Referee) · 27 May 2017

This paper discusses an interesting modelling approach, in which two different models (vegetation and fish) are combined. The main argument to do so would be that these two different species/agents require different flow regimes and that they influence each other. My comments are mainly focussing on the way the argument is presented.

Introduction

If I understand the argument correctly, the claim is that environmental flows for fish are typically studied. Apparently, "longer term species" are neglected. Are these longer-term species the vegetation? If so, this should be made much more explicitly.

Assuming that the paper focusses on fish and vegetation, the introduction should clarify much more how these two relate to each other. Riparian restoration, for example, is discussed without any clear relation to the life cycles of fish (and other species).

The research questions are not defined in a way that allows any other answer than that excluding habitat analysis is wrong. The second question introduces the term "overlook" which suggests that the researchers already know the answer to question 1. Then why pose it?

What is the "structural response" of riparian vegetation?

Detailed remarks: page 1, line 33: I would avoid using words like "truly". page 1, line 36: why "Therefore"? page 2, line 6: "It is now in agreement" with only one reference is not very strong. page 2, lines 10-11: what does "holistic"mean? Why are these drafted in these two countries, what to they entail? page 2, line 15: why "clearly"? page 2, line 16: do you need the word "biased"? page 2, line 35: "In what extent", does that exist?

Methods

It is clear how the models link to the measurements. It is not clear at all how the models have been calibrated etcetera. The whole paper does not discuss sensitivity or similar concepts. Can we be sure that the model results are similar to the data for the right reason? Please explain.

Results

Detailed remarks: Page 7, lines 19-35: please separate in three paragraphs. The reader can hardly distinguish between the three cases. Page 8, lines 7-17: when the term "significant" is introduced, I would recommend to use it more specific and add "statistically" (as is finally done on page 9, line 27). I would also recommend including the numbers here and not refer to annexes too soon.

Discussion

Lines 23-26 on page 9 seem to be rather important. I would suggest that these could be more prominent. Morphodynamics and fish need to be understood together. This is done by first modelling vegetation and than fish. I have some questions on that process that could be taken on board in the discussion? - Is the one-way relation that is modelled (from vegetation to fish) possibly a two-way relation (back from fish to vegetation)? - Does the modelling assume stability of the riverbed and -shape, apart from vegetation? - If so, is that a problem? - How do the modelling uncertainties of the two models relate and introduced in each other?

Detailed remarks: Page 9, line 16: what does "pushed through" mean? Page 9, line 36: why suddenly the term "substantially"?

Other remarks:

Please check the abstract. The second sentence is very difficult to understand. The numbers mentioned do not easily relate to numbers that are discussed in the main text.

The language needs to be improved. For example, several times the word "inputted" is used, which as far as I know does not exist. Proofreading may be recommended.

I am in agreement with the comments of my colleague, who provided much more detail on which parts of the paper should be improved and clarified.

---

## Author Comment (AC1) · 8 Jun 2017

Reviewer 1

*Main concerns:*

*1. The methodological section should include some clarifications about the structure of…*

The authors agree that a figure with a methodology scheme would improve the understanding of the modeling procedures. Such figure will be provided and included in the manuscript or added to the supplementary material according to the decision of the HESSD handling editor. The data transference from one model to another goes like this: the CASiMiR-vegetation model provides the riparian landscape scenarios resulting from each flow regime; these landscape scenarios, discriminated by succession phases, are transformed into roughness maps that are inputted into the River2D model and will characterize the channel roughness of each scenario during the hydrodynamic modeling for the corresponding flow regime. This explanation can be found in page 6, from line 22 to 29 but will be strengthened by the methodology scheme. Also, an additional paragraph stating strengths and limitations of the models will be included in the methods section.

About the aquatic zone, this is a misinterpretation resulting from different aquatic zone concepts and the authors realize now that such a simple explanation as the one presented in page 6, line 38, may go unnoticed or be misunderstood by a reader unfamiliar with CASiMiR-vegetation model. By aquatic zone, the authors were not talking about the channel wetted area, which is variable throughout the year. The aquatic zone in the sense of CASiMiR-vegetation model is the permanently inundated area by the river during the hydrologic year, this is, the area flooded by the absolute minimum discharge of the river. The concept underlying the definition of this zone is that herein riparian vegetation is not capable of establishing and develop because it is always under water and riparian vegetation needs grounds that are at least in some parts of the hydrological year out of water. This is only a concept that is incorporated in the modeling of the ecological succession of riparian vegetation by CASiMiR-vegetation in order to save computational resources that would be used in modeling areas that you know will never have riparian vegetation (as long as this area is permanently inundated). When using the hydrodynamic model River2D, all the river stretch is considered and the channel roughness is set according to the succession phases of each riparian landscape scenario as well as by the river bed substrate where riparian vegetation is determined to be inexistent (this is, the aquatic zone *sensu* CASiMiR-vegetation).

The riparian vegetation landscape resulting from the CASiMiR-vegetation model will interact with river flow because the discharges in the considered flow regimes are always greater than the minimum discharge considered for the aquatic zone defined in the CASiMiR-vegetation model. Accordingly, all the area submerged by river flow in addition to this aquatic zone in the context of CASiMiR-vegetation will directly run through some succession phase of riparian vegetation. Furthermore, the interaction between river flow and riparian vegetation in the margins will influence the overall hydraulics, due to flow deflection or water retention in the margins, for instance, and thus, also the hydraulic parameters in the area without riparian vegetation will be affected. A better explanation about the definition of aquatic zone considered in the CASiMiR-vegetation model will be included in the text.

*2. About flow regime definition (section 2.3) authors mentioned that…*

The environmental flow regimes considered in this study were created in Ferreira et al. (2014) and used here. The proposal for an environmental flow regime created in Ferreira et al. (2014)

considered two different flow regime components: a monthly flow regime addressing fish requirements and a multiannual flow regime composed by floods with different recurrence intervals addressing riparian vegetation requirements. The first component of this environmental flow regime, i.e., the flow regime addressing fish requirements (named Eflow in the manuscript) was determined according to the Instream Flow Incremental Methodology. The second component of this environmental flow regime (floods with a certain recurrence interval) was determined according to Rivaes et al. (2015). The environmental flow regimes used in this study were considered as an adaptation from Ferreira et al. (2014) because the authors used just the fish-addressing component as the standard procedure of an environmental flow regime considering only fish requirements (Eflow) and another environmental flow regime addressing fish and riparian requirements (named Eflow&Flush in the manuscript) composed by both components of the environmental flow regime proposed in Ferreira et al. (2014). Sentences will be rewritten for a better understanding.

**3. Regarding environmental flows considering riparian vegetation…**

In this study, the sediment transport originated by the environmental flow regimes was not considered. The authors chose this approach based on their expert knowledge in previous studies, (namely, in Rivaes et al., 2015), where the sediment transport caused by dam flood discharges were modeled in two case studies and where results demonstrated, in both cases, that such flood discharges were not relevant for river bed degradation. Furthermore, in rivers with a bed substrate of much smaller sizes (pebbles and sand). As requested by the reviewer, a paragraph will be included discussing this approach in the discussion section.

**4. Regarding vegetation modelling, CASiMiR model lacks of a crucial process such as the morphological evolution of the river…**

The CASiMiR-vegetation model does not uses a fixed topography. CASiMiR-vegetation is not a hydraulic model but topography can be updated on a yearly basis during the input data upload into the dynamic module (see figure 21 of the CASiMiR-vegetation manual, page 35) of the model. Therefore, a comparison between modeling runs using fixed and variable topographies is possible using the CASiMiR-vegetation model. Nevertheless, the authors totally agree with the reviewer and are well aware of this interaction between river morphodynamics and riparian vegetation with bi-directional influences, which is particularly important in very morphodynamic rivers. Although, the references provided by the reviewer are not good examples as those only refer to gravel riverbeds, which is not the case of our study sites. In fact, as mentioned by Corenblit et al. (2011), research on the temporal scales of geomorphic and ecological processes is still scarce, even more for such coarse substrate rivers. Every case must be analyzed with a critical thinking. In this case, using a fixed topography may be considered a flaw when modeling riparian vegetation but the authors made it intentionally. By using a fixed topography, the authors were able to isolate and better analyze the effect of riparian landscape degradation on river hydraulics. Incorporating topography changes in the modeling runs would not allow to address the results to a solely factor. The reasons that lead the authors to consider a fixed topography during this 10 year period were: 1- the typical substrate of both study sites is armored and very coarse (boulders, large boulders and bedrock), as mentioned; 2- no significant differences were found during the substrate analysis of the different succession phases regarding the data collected in the field survey that could allow the authors to infer substrate and topographic changes according to the succession phase, and therefore authors agreed not to forecast morphological changes in observed fairly stable topographies; 3- previous studies of the authors regarding this matter show that the considered floods do not bring substantial

changes to river geomorphology; 4- flow velocities and water depths experienced in the study sites for monthly discharges are not expected to induce erosion in the existing river bed; 5- the study sites are located in a fairly steep valley in which the river is not allowed to meander considerably during such a short time scale; 6- this work is on a first part focused on the modeling of riparian vegetation dynamics in a representative proportion of the existing river landscape features and although the position of these features can eventually change over time, their overall proportion is expected to remain constant (Stanford et al., 2005) and posing no noteworthy effects on the analysis of vegetation dynamics. In fact, this last reason was the basis of the modeling methodology used by Politti et al. (2014) in which they verified that only from 25 years onwards the difference in the results of riparian vegetation landscapes using a fixed topography became notable in some parts of the study site. This was possible to observe because different topographies of the study site were available. Indeed, this study was conducted for the purpose of analyzing the effects of climate change on the riparian vegetation in an Alpine river exposed to a greater morphodynamics but provides support for the decision of the authors in disregarding morphodynamics in a minor time period and for a much more stable river. Furthermore, one must not forget that in this particular case, the model calibration and validation results while using the same methodology exhibited a good agreement with the observed riparian landscape. Thus, considering the previous premises, the authors are confident that the disregard of the river morphodynamics in this case does not bring a tangible shortcoming to this research. Notwithstanding, in order to clarify the reasoning for using a fixed topography, the authors will include a paragraph in the methods chapter to explain better the use of this approach and another in the discussion section debating this option in the analysis.

*5. About results presentation, now this section is a bit confuse and I think it will benefit from…*

The authors agree that one or two sentences can be included in the results section summarizing the main results. Although the authors provide the response to the research questions in the manuscript, text will be rewritten to explicitly respond to those questions.

*Specific comments:*

*Title: As your study encompass a decade, talk about "the long-term" is not very appropriate…*

This "long-term" expression refers to the efficiency of environmental flows assessed by habitat modeling methods on the aquatic biota for which requirements it is said these flows are addressed to. The focus of this study is not on the effects of flow regulation on riparian vegetation but on the effects of environmental flows on aquatic fauna, surrogated by microhabitat metrics, in reaches for which environmental flow prescriptions are settle considering only aquatic fauna requirements. Accordingly, this research is more of a microhabitat analysis in which authors analyze the influence of the riparian landscape degradation on the hydraulic parameters water depth and flow velocity. Hence, for this spatial scale, the appropriate time scale would be, according to Frissell et al. (1986), of about $10^{-1}$ to at most 10 years. The authors considered a time frame of a decade in order to obtain a notable response of the riparian landscape to flow regulation without the geomorphology constrains discussed previously, which in fact revealed to be appropriate to disclose a significant trend in the riparian landscape response, but the focus is still on the effects of microhabitat amendments for fishes, which clearly change gradually every year until the end of the decade. Indubitably, considering that dams are built to last a century or more, those amendments will certainly continue to happen until a metastable state equilibrium occurs over time. In this sense, we are talking about the influence of environmental flows obtained by habitat modeling methods over

the long-term perspective of the aquatic microhabitat. The authors propose to change the title to: "Importance of considering riparian vegetation requirements for the long-term efficiency of environmental flows on aquatic microhabitat".

*Introduction: Introduction section provides an appropriate "stat-of-the-art" about…*

As responded before, the authors will provide clearer answers to the research questions.

*Methods: Study site: Page 3 line 9: Authors use a very general reference…*

Information about discharge data and return periods will be included in the description of the study area. Figure 1 will be changed accordingly to the reviewer comments.

*Data collection: Please, give a brief description about field procedures like…*

Although sent to literature to keep the manuscript not too long, a brief description about field procedures will be included in the methods section.

*Riparian vegetation modelling Page 5 lines 36-39: "the hydrological regime is inputted…*

Yes, the hydric stress imposed by the duration of extreme low flows is also accounted by CASiMiR-vegetation model. The magnitude and duration of extreme low flows are reflected in the mean annual discharge of the river, which is a model input used as a reference for the general habitat conditions that determine the expected riparian landscape according to the thresholds of riparian succession phases. This information was slightly approached in the text (page 6, lines 1-2) but this paragraph will be rewritten for a better explanation.

*Page 6 Line 8: Authors have included many supplementary material which is very appreciate…*

Will be done according to the reviewer suggestions.

*Page 6 Lines 12-13: "The resulting riparian… hereafter named natural, Eflow and Eflow&Flush…*

The authors propose to change the word "habitats" by "landscape".

*Table S5: in supplementary material contains some of the vegetation model parameters…*

The authors can only speculate about the reasons for this discrepancy as no in-depth research was conducted to ascertain this issue. The resistance thresholds of riparian vegetation to shear stress deeply rely on the river geomorphology and ecophysiological traits of the riparian species. Differences between Politti et al. (2014) and these case studies are found in river type, flow regime, geomorphology, hydraulics and riparian species. The Austrian case study is located in an Alpine river of much greater dimension than the considered Mediterranean case, with greater catchment area, higher and longer maximum discharges, longer flood durations and with a phenomenon known as glacial milk, which confers much more sediment load to the flowing water. Furthermore, the higher discharges in this Alpine river occur during summer, when vegetation is in its vegetative period and consequently more vulnerable to these stresses. On the other hand, Mediterranean species are well adapted to the flow regimes flashiness, characterized by very short flood durations, mostly occurring out of its vegetative period. These are all differences in the river systems that can explain the different calibration parameterization of riparian vegetation resistance thresholds in these two river systems.

*Hydrodynamic modeling Page 6 Lines 29-31 31. "The hydraulic characteristics of each habitat…*

Yes, at this stage the hydraulic parameter values were considered all together regardless from succession phases. As mentioned in the manuscript research questions (page 2 lines 30 to 38), one of the objectives of this research was to question the capacity of fish-addressed environmental flows in maintaining fish habitat availability in the long-term. The used approach was successful in this task, as the considered hydraulic parameters water depth and flow velocity were significantly different between scenarios. An analysis of these hydraulic parameters by succession phase is feasible but would not bring (in this case) substantial increase of information as the main succession phase interacting with river flow is Early Succession Woodland phase. This is due to the low discharges considered in the Eflow. Besides, one may not forget that the water depth and flow velocity in a certain microhabitat do not result only from the existing local conditions, but also from surrounding conditions. Furthermore, this kind of analysis would require data that the authors do not have, such as, fish preference curves for each type of vegetation indicator of each succession phase or the preference of fish species for hanging vegetation, for instance. For these reasons the authors think that analyzing the use of fish by each succession phase is quite out of the scope of the paper.

*Results: In general, this section could be better structured with some sub-sections…*

The authors agree and sub-sections will be included in the text.

*Page 7 Line 19: Here authors use "habitat" in a different context…*

The authors agree and will proceed as proposed to the previous comment regarding this matter.

*Page 7 Line 36: "The changes undertaken by the riparian vegetation facing different flow regimes are able…*

The roughness values refer to the entire study site areas in each scenario while the values of water depth and flow velocity only refer to the areas inundated by the considered discharges as only there one can find water depth and flow velocity estimates. Once again, the authors are not analyzing the habitat suitability according with fish preference for the type of vegetation, but according to the preference of fish for water depth, flow velocity and substrate. The habitat suitability regarding these parameters can be computed independently from the type of vegetation. Although, the type of vegetation interfere in these parameters due to different characteristic roughness, which were considered during the hydrodynamic modeling.

*Page 8 Lines 3-6: Comments about ks, it is not clear which comment refers to figure 4…*

Figure 4 shows the distribution of the values regarding the considered hydraulic parameters for each study site. Due to the great amount of data, differences between landscapes are not very noticeable in some cases. Consequently, the authors decided to include in the supplementary material the tests results that support the author statements regarding the significant differences in the hydraulic characteristics of each riparian landscape. Additional clarifications will be included in the text.

*Figure 5: It is not indicated to which reach refer each set of graphics…*

This information will be added to the figure caption.

*Page 8 Line 34: "The Eflow habitat consistently provides less habitat suitability during autumn…*

This is an error. You should read "… nase juveniles and adults…". Text will be corrected.

*Discussion: As I mentioned before, the modelling techniques that authors used have…*

As mentioned previously, limitations of the used techniques will be discussed in the discussion section. The used terms will be homogenized and the paragraph rewritten.

Reviewer 2

*Introduction*

*If I understand the argument correctly, the claim is that environmental flows for fish are…*

When the authors refer to species with longer lifecycles they are talking in general to mention that the usual approach on environmental flows only takes into consideration the intra-annual variability of the fluvial system but there is an inter-annual variation that can influence the life cycle of many organisms and which should be considered. In this case, the authors used riparian vegetation as the example for their case study. Riparian vegetation was then introduced in the next paragraph (page 2, lines 24 and 25), where the authors explain its connection to the flow regime and to the aquatic fauna when one mentions that riparian vegetation has a clear significance in the habitat improvement of aquatic systems, with references provided. Notwithstanding, riparian vegetation will be mentioned as example in the considered sentence.

*Assuming that the paper focusses on fish and vegetation, the introduction should clarify much…*

Because of the manuscript length, particularly the introduction section, the authors did not find necessary to present a deep bibliographic review about the relations between riparian vegetation and aquatic fauna. The reason for this decision is that the main the scope of the manuscript is on the efficiency of environmental flows when ignoring riparian requirements and fish species were used as a surrogate for the response of aquatic communities to the expected riparian landscape changes. This response was approached from an ecohydraulic point of view, in which the river hydraulics was the considered linkage between these two communities (page 2, lines 30-33).  Even so, references were presented showing the influence of riparian communities on aquatic species assemblages in order to highlight the importance of restoring riparian vegetation not only for the improvement of these communities but also for the inherent improvements that such restoration can bring to aquatic species (page 2, line 26). Nevertheless, this paragraph will be improved to better clarify how riparian vegetation and fish species relate to each other.

*The research questions are not defined in a way that allows any other answer than that…*

If the authors understood correctly the concern of reviewer 2, the objective of this research is not to demonstrate that habitat modeling is the only good method for environmental flow assessment. In fact, the authors never make that statement throughout the manuscript. However, this study is based on habitat analysis and the authors needed to use habitat modeling to address their research questions. The term "overlook" of the second research question means the disregard that common approaches for environmental flows definition have for other biological communities. This comes from the bibliographic review in the introduction section where it is mentioned the need for environmental flows to address the ecological requirements of different biological communities rather than only a single biological group, which usually are always fish species. Based on the bibliographic review, the authors did not know the answer to

question 1 but they raised their hypotheses. First, the authors question if the fish habitat would stay the same throughout the years facing the degradation of riparian vegetation due to flow regulation. If changes in habitat are noticeable, second question is: what is the extent of that change due to neglecting riparian requirements. Nonetheless, the authors understand the concern of the reviewer and this sentence will be changed by a more consensual one with the same meaning.

*What is the "structural response" of riparian vegetation?*

Riparian vegetation is structured or arrayed in space and time along gradients in the three river dimensions: longitudinal, lateral and vertical. A response of riparian vegetation to a certain driver implying a change in this structure is denominated a structural response of riparian vegetation. This expression is widely used in vegetation science nomenclature (e.g. NRC, 2002; Naiman et al., 2005), and the authors will provide such references on the text.

*Detailed remarks:*

*Page 1, line 33: would avoid using words like "truly".*

This word will be changed by "Actually".

*Page 1, line 36: why "Therefore"?*

The authors mean that this is a consequence of the previous sentence. However, this will be changed for a better understanding.

*Page 2, line 6: "It is now in agreement" with only one reference is not very strong.*

The authors agree. More references will be added, namely, Brisbane Declaration (2007); Arthington (2012); Poff et al. (1997); Acreman et al. (2014); Acreman and Ferguson (2010) and Davis and Hirji (2003).

*Page 2, lines 10-11: what does "holistic"mean?*

There are different methodology types for environmental flow assessment. One of those types are named "holistic methodologies" (see Arthington et al., 2003; Dyson et al., 2003; Tharme, 2003; Arthington, 2012) also known as "function analysis" (see Dyson et al., 2003). Holistic methodologies are meant to address river systems as a whole. These methodologies emerged parallel in Australia and South Africa and share one same purpose: to protect or restore the flow-related biophysical components and ecological processes of the entire river system. This term will be followed by such references to support its context.

*Page 2, line 15: why "clearly"?*

The authors state "clearly" supported in the systematic synthesis of the global literature regarding environmental flows done by Gillespie et al. (2014), which realized that the majority of the studies reported to fish response and given the importance of all trophic levels in sustaining freshwater ecological integrity, the predisposition towards monitoring of this traditional indicator taxa is a concern. According to this author, also Olden et al. (2014) found this tendency and therefore verified a clear need for diversification of monitoring strategies to cover less typically monitored taxa in future studies. Notwithstanding, this word "clearly" will be removed without changing the meaning of the phrase.

*Page 2, line 16: do you need the word "biased"?*

This comes in line with the previous question about whether environmental flow assessments are prone to fish response evaluation rather than other biological groups. This word will be removed.

*Page 2, line 35: "In what extent", does that exist?*

The authors propose to change to "to what extent".

*Methods*

*It is clear how the models link to the measurements. It is not clear at all how the models have…*

The calibration and validation of the models were referred to existing literature in order to control the length of the manuscript (see page 6, lines 3-6 and 19-20). Since the scope of the manuscript is not to present a particular model, the exhaustive description of the calibration methodology was not deemed necessary as plenty published papers already describe thoroughly the validation methodology of these models. Notwithstanding, the authors referred the calibration methodology employed in this case and presented the result of such calibration, so the reader can verify the accuracy of the model in this specific case study. Cohen's Kappa statistic was the chosen measure to evaluate the calibration of the CASiMiR-vegetation model because this is considered a good measure to analyze this model's accuracy and the most often used measure of inter-rater agreement for categorical classifications. Furthermore, it has an advantage over sensitivity, since it corrects the overall accuracy of model predictions by compensating for random agreement. Considering the River2D model, the authors first estimated the bed roughness coefficient, the roughness height, in accordance with the observations of bed material and bedform size for the natural flow regime. The final values of roughness height were obtained by calibrating the water surface elevation measured in different cross-sections in the field and the model results. For the different scenarios (i.e. Eflow regime and Eflow&Flush regime) the roughness height values were changed according to the expected riparian vegetation maps. In the end, the employed models are widely used and scientifically accepted tools that were calibrated for the study sites according to recognized methodologies. Calibration results were analyzed by comparison to observed data and achieved a good classification according to different categorizations of map classification agreement. All of these provide confidence to the authors that the model results are right and simulate correctly the considered fluvial system.

*Results*

*Detailed remarks:*

All these remarks will be addressed.

*Discussion*

*Lines 23-26 on page 9 seem to be rather important. I would suggest that these could…*

Lines 23-26 will be stated in a more prominent way. The two-way relation (back from fish to vegetation) is not considered in this modeling work. The two models employed in this study do not consider the effect of fish on vegetation or morphodynamics. The authors do not think this is applicable considering the river particle dimensions. River bed was considered stable during modeling runs (please see the response to reviewer 1 regarding this matter). The topics

mentioned for this section by the reviewer 2 are pertinent and will be discussed in the discussion section.

*Page 9, line 16: what does "pushed through" mean?*

This means that such approach puts forward an ecological modeling procedure that is more realistic than the actual paradigm in the assessment of environmental flows by means of fish habitat modeling. This expression will be changed by "enables".

*Page 9, line 36: why suddenly the term "substantially"?*

This means that the habitat availability originated by the Eflow changes a lot when compared to the natural and Eflow&Flush flow regimes. This expression will be changed to "noticeably".

*Other remarks:*

*Please check the abstract. The second sentence is very difficult to understand…*

The second sentence of the abstract will be modified for a better understanding. The numbers mentioned in the abstract were introduced in a way that the authors thought to be more comprehensive and appealing to the reader without reading the entire article.

*The language needs to be improved. For example, several times the word "inputted" is…*

The manuscript was English revised by Elsevier Language Editing Services prior to the submission to HESSD journal and holds a certificate from this institution. The word "inputted" is the past tense and past participle of the verb "input". This can be found in different English dictionaries, like the Cambridge dictionary (http://dictionary.cambridge.org/dictionary/english/input) or the Oxford dictionary (https://en.oxforddictionaries.com/definition/input). If the reviewers and/or the handling editor still require a more thorough proofreading, the authors can readdress this issue to the Elsevier Language Editing Services in order to meet the expectations of the reviewers and handling editor.

Reference list:

Acreman, M. C., and Ferguson, J. D.: Environmental flows and the European Water Framework Directive, Freshwater Biology, 55, 32-48, 10.1111/j.1365-2427.2009.02181.x, 2010.

Acreman, M. C., Overton, I. C., King, J., Wood, P. J., Cowx, I. G., Dunbar, M. J., Kendy, E., and Young, W. J.: The changing role of ecohydrological science in guiding environmental flows, Hydrological Sciences Journal, 59, 433-450, 10.1080/02626667.2014.886019, 2014.

Arthington, A.H.: *Environmental flows: saving rivers in the third millennium.* Univ of California Press, 2012

Arthington, A.H., Tharme, R.E., Brizga, S.O., Pusey, B.J., and Kennard, M.J.: "Environmental flow assessment with emphasis on holistic methodologies", in: *Second International Symposium on the Management of Large Rivers for Fisheries*), 37-66, 2013.

Brisbane Declaration: The Brisbane Declaration. Environmental flows are essential for freshwater ecosystem health and human well-being, Declaration of the 10th International River*symposium* and International Environmental Flows Conference, Brisbane, AUS, 2007.

Corenblit, D., Baas, A.C.W., Bornette, G., Darrozes, J., Delmotte, S., Francis, R.A., et al.: Feedbacks between geomorphology and biota controlling Earth surface processes and landforms: A review of foundation concepts and current understandings. *Earth-Science Reviews* 106**,** 307-331. doi: http://dx.doi.org/10.1016/j.earscirev.2011.03.002, 2011.

Davis, R., and Hirji, R.: Environmental flows: concepts and methods. Water Resources and Environment Technical Note nº C1, Environmental Flow Assessment series, World Bank, Washington, DC, 27 pp., 2003.

Dyson, M., Bergkamp, G., and Scanion, J. (eds.).: *Flow. The Essentials of Environmental Flows.* Gland, Switzerland and Cambridge, UK: IUCN, 2003.

Ferreira, M.T., Pinheiro, A.N., Santos, J.M., Boavida, I., Rivaes, R., and Branco, P.: "Determinação de um regime de caudais ecológicos a jusante do empreendimento de Alvito". (Lisboa: Instituto Superior de Agronomia, Universidade de Lisboa). Relatório Final. Estudo realizado pelo Instituto Superior de Agronomia para a ATKINS, no âmbito do Protocolo de Colaboração ATKINS – ISA-ADISA, de 19 de Junho de 2012 (URL: http://hdl.handle.net/10400.5/13341), 2014.

Frissell, C., Liss, W., Warren, C., and Hurley, M.: A hierarchical framework for stream habitat classification: Viewing streams in a watershed context. *Environmental Management* 10(2)**,** 199-214, 1986.

Gillespie, B.R., Desmet, S., Kay, P., Tillotson, M.R., and Brown, L.E.: A critical analysis of regulated river ecosystem responses to managed environmental flows from reservoirs. *Freshwater Biology* 60(2)**,** 410-425. doi: 10.1111/fwb.12506, 2014.

Naiman, R.J., Décamps, H., and McClain, M.E. (eds.): *Riparia - Ecology, conservation and management of streamside communities.* London, UK: Elsevier academic press, 2005.

NRC, N.R.C.: *Riparian Areas: Functions and Strategies for Management.* Washington, D.C., USA: The National Academies Press, 2002.

Olden, J.D., Konrad, C.P., Melis, T.S., Kennard, M.J., Freeman, M.C., Mims, M.C., et al.: Are large-scale flow experiments informing the science and management of freshwater ecosystems? *Frontiers in Ecology and the Environment* 12(3)**,** 176-185. doi: 10.1890/130076, 2014.

Poff, L. N., Allan, J. D., Bain, M. B., Karr, J. R., Prestegaard, K. L., Richter, B. D., Sparks, R. E., and Stromberg, J. C.: The natural flow regime, Bioscience, 47, 769-784, 1997.

Rivaes, R., Rodríguez-González, P.M., Albuquerque, A., Pinheiro, A.N., Egger, G., and Ferreira, M.T.: Reducing river regulation effects on riparian vegetation using flushing flow regimes. *Ecological Engineering* 81**,** 428-438. doi: 10.1016/j.ecoleng.2015.04.059, 2015.

Stanford, J.A., Lorang, M.S., and Hauer, F.R.: The shifting habitat mosaic of river ecosystems. *Verh. Internat. Verein. Limnol.* 29. doi: 0368-0770/05/1940-0013 $ 2.00, 2005.

Tharme, R.E.: A global perspective on environmental flow assessment: emerging trends in the development and application of environmental flow methodologies for rivers. *River Research and Applications* 19(5-6)**,** 397-441. doi: 10.1002/rra.736, 2003.

---

## Author Response (AR1)

The authors are very grateful for the comments of the anonymous reviewer 1 and reviewer 2 – Maurits Ertsen. The concerns of the reviewers are pertinent and their suggestions much appreciated. The authors believe that such constructive criticism certainly improved the quality of the paper. An extensive revision was made with clarification of the issues raised by the editor and referees. Particularly, we improved the methods section and included a figure summarizing the methodology, we provided a detailed examination about our option for considering a static geomorphology during modeling, and we deposited all data in a reliable public repository. We believe that this revised version is much improved and we hope that that it now satisfies the concerns of the referees and handling editor. Bellow you will find detailed responses to the reviewers comments (in blue), stating exactly the changes performed into the revised manuscript. Thank you for allowing us the opportunity to revise this manuscript and contribute to Hydrology and Earth System Sciences Journal. Please do not hesitate to contact me if you have any questions.

**Reviewer 1**

**Main concerns:**

**1. The methodological section should include some clarifications about the structure of...**

The authors agree that a figure with a methodology scheme would improve the understanding of the modeling procedures. Such figure was included in the manuscript. Furthermore, the methodological section was improved by adding further details to ensure reproducibility of the results. In detail, the data transference from one model to another goes like this: the CASiMiR-vegetation model provides the riparian landscape scenarios resulting from each flow regime; these landscape scenarios, discriminated by succession phases, are transformed into roughness maps that are inputted into the River2D model and will characterize the channel roughness of each scenario during the hydrodynamic modeling for the corresponding flow regime. This explanation can be found in page 6, from line 22 to 29 but was strengthened by the methodology scheme and by the added details into the methods section. Also, an additional paragraph stating strengths and limitations of the models was included in the methods section (now in P7, L5-9; P8, L22-25).

About the aquatic zone, this is a misinterpretation resulting from different aquatic zone concepts and the authors realize now that such a simple explanation as the one presented in page 6, line 38, may go unnoticed or be misunderstood by a reader unfamiliar with CASiMiR-vegetation model. By aquatic zone, the authors were not talking about the channel wetted area, which is variable throughout the year. The aquatic zone in the sense of CASiMiR-vegetation model is the permanently inundated area by the river during the hydrologic year, this is, the area flooded by the absolute minimum discharge of the river. The concept underlying the definition of this zone is that herein riparian vegetation is not capable of establishing and develop because it is always under water and riparian vegetation needs grounds that are at least in some parts of the hydrological year out of water. This is only a concept that is incorporated in the modeling of the ecological succession of riparian vegetation by CASiMiR-vegetation in order to save computational resources that would be used in modeling areas that you know will never have riparian vegetation (as long as this area is permanently inundated). When using the hydrolynamic model River2D, all the river stretch is considered and the channel roughness is set according to the succession phases of each riparian landscape scenario as well as by the river

bed substrate where riparian vegetation is determined to be inexistent (this is, the aquatic zone *sensu* CASiMiR-vegetation).

The riparian vegetation landscape resulting from the CASiMiR-vegetation model will interact with river flow because the discharges in the considered flow regimes are always greater than the minimum discharge considered for the aquatic zone defined in the CASiMiR-vegetation model. Accordingly, all the area submerged by river flow in addition to this aquatic zone in the context of CASiMiR-vegetation will directly run through some succession phase of riparian vegetation. Furthermore, the interaction between river flow and riparian vegetation in the margins will influence the overall hydraulics, due to flow deflection or water retention in the margins, for instance, and thus, also the hydraulic parameters in the area without riparian vegetation will be affected. A better explanation about the definition of aquatic zone considered in the CASiMiR-vegetation model was also included in the text (now in P9, L13-16).

**2. About flow regime definition (section 2.3) authors mentioned that...**

The environmental flow regimes considered in this study were created in Ferreira et al. (2014) and used here. The proposal for an environmental flow regime created in Ferreira et al. (2014) considered two different flow regime components: a monthly flow regime addressing fish requirements and a multiannual flow regime composed by floods with different recurrence intervals addressing riparian vegetation requirements. The first component of this environmental flow regime, i.e., the flow regime addressing fish requirements (named Eflow in the manuscript) was determined according to the Instream Flow Incremental Methodology. The second component of this environmental flow regime to Rivaes et al. (2015). The environmental flow regimes used in this study were considered as an adaptation from Ferreira et al. (2014) because the authors used just the fish-addressing component as the standard procedure of an environmental flow regime considering only fish requirements (Eflow) and another environmental flow regime addressing fish and riparian requirements (named Eflow&Flush in the manuscript) composed by both components of the environmental flow regime proposed in Ferreira et al. (2014). Sentences were rewritten for a better understanding.

**3. Regarding environmental flows considering riparian vegetation...**

In this study, the sediment transport originated by the environmental flow regimes was not considered. The authors chose this approach based on their expert knowledge in previous studies, (namely, in Rivaes et al., 2015), where the sediment transport caused by dam flood discharges were modeled in two case studies and where results demonstrated, in both cases, that such flood discharges were not relevant for river bed degradation. Furthermore, in rivers with a bed substrate of much smaller sizes (pebbles and sand). As requested by the reviewer, a paragraph was included discussing this approach in the discussion section (now P13, L30-34).

**4. Regarding vegetation modelling, CASiMiR model lacks of a crucial process such as the morphological evolution of the river...**

The CASiMiR-vegetation model does not uses a fixed topography. CASiMiR-vegetation is not a hydraulic model but topography can be updated on a yearly basis during the input data upload into the dynamic module (see figure 21 of the CASiMiR-vegetation manual, page 35) of the model. Therefore, a comparison between modeling runs using fixed and variable topographies is possible using the CASiMiR-vegetation model. Nevertheless, the authors totally agree with the reviewer and are well aware of this interaction between river morphodynamics and riparian

vegetation with bi-directional influences, which is particularly important in very morphodynamic rivers. Although, the references provided by the reviewer are not good examples as those only refer to gravel riverbeds, which is not the case of our study sites. In fact, as mentioned by Corenblit et al. (2011), research on the temporal scales of geomorphic and ecological processes is still scarce, even more for such coarse substrate rivers. Every case must be analyzed with a critical thinking. In this case, using a fixed topography may be considered a flaw when modeling riparian vegetation but the authors made it intentionally. By using a fixed topography, the authors were able to isolate and better analyze the effect of riparian landscape degradation on river hydraulics. Furthermore, one may not forget that the authors already tested such flow regimes in other study sites with greater morphodynamics and showed that these flow regimes will not change topography in more than a few centimeters in a decade (see Rivaes et al., 2015). Also, incorporating topography changes in the modeling runs would not allow to address the results to a solely factor. The reasons that lead the authors to consider a fixed topography during this 10 year period were: 1- the typical substrate of both study sites is armored and very coarse (boulders, large boulders and bedrock), as mentioned; 2- no significant differences were found during the substrate analysis of the different succession phases regarding the data collected in the field survey that could allow the authors to infer substrate and topographic changes according to the succession phase, and therefore authors agreed not to forecast morphological changes in observed fairly stable topographies; 3- previous studies of the authors regarding this matter show that the considered floods do not bring substantial changes to river geomorphology; 4- flow velocities and water depths experienced in the study sites for monthly discharges are not expected to induce erosion in the existing river bed; 5- the study sites are located in a fairly steep valley in which the river is not allowed to meander considerably during such a short time scale; 6- this work is on a first part focused on the modeling of riparian vegetation dynamics in a representative proportion of the existing river landscape features and although the position of these features can eventually change over time, their overall proportion is expected to remain constant (Stanford et al., 2005) and posing no noteworthy effects on the analysis of vegetation dynamics. In fact, this last reason was the basis of the modeling methodology used by Politti et al. (2014) in which they verified that only from 25 years onwards the difference in the results of riparian vegetation landscapes using a fixed topography became notable in some parts of the study site. This was possible to observe because different topographies of the study site were available. Indeed, this study was conducted for the purpose of analyzing the effects of climate change on the riparian vegetation in an Alpine river exposed to a greater morphodynamics but provides support for the decision of the authors in disregarding morphodynamics in a minor time period and for a much more stable river. Furthermore, one must not forget that in this particular case, the model calibration and validation results while using the same methodology exhibited a good agreement with the observed riparian landscape. Thus, considering the previous premises, the authors are confident that the disregard of the river morphodynamics in this case does not bring a tangible shortcoming to this research. Notwithstanding, in order to clarify the reasoning for using a fixed topography, the authors included a paragraph in the methods chapter to explain better the use of this approach and another in the discussion section debating this option in the analysis (now in P8, L1-13 and P13, L21-30).

**5. About results presentation, now this section is a bit confuse and I think it will benefit from...**

The authors agree that one or two sentences can be included in the results section summarizing the main results. Although the authors provide the response to the research questions in the

manuscript, sentences were included to explicitly respond to those questions (now in P12, L4-6; P13, L1-6).

**Specific comments:**

**Title: As your study encompass a decade, talk about "the long-term" is not very appropriate...**

This "long-term" expression refers to the efficiency of environmental flows assessed by habitat modeling methods on the aquatic biota for which requirements it is said these flows are addressed to. The focus of this study is not on the effects of flow regulation on riparian vegetation but on the effects of environmental flows on aquatic fauna, surrogated by microhabitat metrics, in reaches for which environmental flow prescriptions are settle considering only aquatic fauna requirements. Accordingly, this research is more of a microhabitat analysis in which authors analyze the influence of the riparian landscape degradation on the hydraulic parameters water depth and flow velocity. Hence, for this spatial scale, the appropriate time scale would be, according to Frissell et al. (1986), of about 10-1 to at most 10 years. The authors considered a time frame of a decade in order to obtain a notable response of the riparian landscape to flow regulation without the geomorphology constrains discussed previously, which in fact revealed to be appropriate to disclose a significant trend in the riparian landscape response, but the focus is still on the effects of microhabitat amendments for fishes, which clearly change gradually every year until the end of the decade. Indubitably, considering that dams are built to last a century or more, those amendments will certainly continue to happen until a metastable state equilibrium occurs over time. In this sense, we are talking about the influence of environmental flows obtained by habitat modeling methods over the long-term perspective of the aquatic microhabitat. The authors propose to change the title to: "Importance of considering riparian vegetation requirements for the long-term efficiency of environmental flows on aquatic microhabitat".

**Introduction: Introduction section provides an appropriate "stat-of-the-art" about...**

Done, the authors provided clearer answers to the research questions (now in P12, L5-6 and P13, L1-6).

**Methods: Study site: Page 3 line 9: Authors use a very general reference...**

Information about discharge data and return periods were included in the description of the study area (now P3, L35-37). Figure 1 was changed according to the reviewer comments.

**Data collection: Please, give a brief description about field procedures like...**

Although sent to literature to keep the manuscript not too long, additional descriptions about field procedures were included in the methods section (now in P4, L36 until P5, L2 and P5, L25-35).

**Riparian vegetation modelling Page 5 lines 36-39: "the hydrological regime is inputted...**

Yes, the hydric stress imposed by the duration of extreme low flows is also accounted by CASiMiR-vegetation model. The magnitude and duration of extreme low flows are reflected in the mean annual discharge of the river, which is a model input used as a reference for the general habitat conditions that determine the expected riparian landscape according to the thresholds of riparian succession phases. This information was slightly approached in the text (page 6, lines 1-2) but an additional paragraph was included for a better explanation regarding this issue (now P7, L17-22).

**Page 6 Line 8: Authors have included many supplementary material which is very appreciate...**

**Done according to the reviewer suggestions.**

**Page 6 Lines 12-13: "The resulting riparian... hereafter named natural, Eflow and Eflow&Flush...**

The authors changed the word "habitats" by "landscape" (throughout the text, e.g. P1, L16). Landscape ecology is the science that studies the interactions between biotic and abiotic structures, functions, and their spatial organization (Zonneveld, 1990). The riparian landscape is therefore a term used in the scope of this science and stands for the specific spatial patterns of riparian vegetation resulting from ecological, geomorphological and hydrological processes. The elements that compose riparian landscapes can be defined as the patches with different vegetation types and succession phases, like the ones that compose our vegetation maps. Thus, riparian landscapes should not be understood here like the river corridor as a landscape, but as the riparian zone functionally dominant feature that contains and connects the elements (Malanson, 1993). Accordingly, we think this term suits well the meaning of what we want to define by riparian landscapes, which is the riparian patch mosaic that derives from a specific flow regime. The explanation of what is a riparian landscape was also included in the text (now in P3, L15-17).

**Table S5: in supplementary material contains some of the vegetation model parameters...**

The authors can only speculate about the reasons for this discrepancy as no in-depth research was conducted to ascertain this issue. The resistance thresholds of riparian vegetation to shear stress deeply rely on the river geomorphology and ecophysiological traits of the riparian species. Differences between Politti et al. (2014) and these case studies are found in river type, flow regime, geomorphology, hydraulics and riparian species. The Austrian case study is located in an Alpine river of much greater dimension than the considered Mediterranean case, with greater catchment area, higher and longer maximum discharges, longer flood durations and with a phenomenon known as glacial milk, which confers much more sediment load to the flowing water. Furthermore, the higher discharges in this Alpine river occur during summer, when vegetation is in its vegetative period and consequently more vulnerable to these stresses. On the other hand, Mediterranean species are well adapted to the flow regimes flashiness, characterized by very short flood durations, mostly occurring out of its vegetative period. These are all differences in the river systems that can explain the different calibration parameterization of riparian vegetation resistance thresholds in these two river systems. For instance, if you look to the Spanish case study presented by García-Arias et al. (2013), in a river stretch much more similar to our case studies, you can see a model parameterization much closer to ours.

**Hydrodynamic modeling Page 6 Lines 29-31 31. "The hydraulic characteristics of each habitat...**

Yes, at this stage the hydraulic parameter values were considered all together regardless from succession phases. As mentioned in the manuscript research questions (page 2 lines 30 to 38), one of the objectives of this research was to question the capacity of fish-addressed environmental flows in maintaining fish habitat availability in the long-term. The used ecohydraulic approach was successful in this task, as the considered hydraulic parameters water depth and flow velocity were significantly different between scenarios. One may not forget that we are assessing fish habitat availability in the light of fish preference for water depth and flow velocity and that we are focused on the influence of riparian landscapes in the river flow patterns. An analysis of these hydraulic parameters by succession phase is feasible but would not bring (in this case) substantial increase of information as the main succession phase

interacting with river flow is Early Succession Woodland phase. This is due to the low discharges considered in the Eflow. Besides, one may not forget that the water depth and flow velocity in a certain microhabitat do not result only from the existing local conditions, but also from surrounding conditions. Furthermore, this kind of analysis would require data that the authors do not have, such as, fish preference curves for each type of vegetation indicator of each succession phase or the preference of fish species for hanging vegetation, for instance. For these reasons the authors think that analyzing the use of fish by each succession phase is quite out of the scope of the paper.

Results: In general, this section could be better structured with some sub-sections...

The authors agree and sub-sections were included in the text.

**Page 7 Line 19: Here authors use "habitat" in a different context...**

The authors agree and proceeded as proposed to the previous comment regarding this matter.

**Page 7 Line 36: "The changes undertaken by the riparian vegetation facing different flow regimes are able...**

The roughness values refer to the entire study site areas in each scenario while the values of water depth and flow velocity only refer to the areas inundated by the considered discharges as only there one can find water depth and flow velocity estimates. Once again, the authors are not analyzing the habitat suitability according with fish preference for the type of vegetation, but according to the preference of fish for water depth, flow velocity and substrate. The habitat suitability regarding these parameters can be computed independently from the type of vegetation. Although, the type of vegetation interfere in these parameters due to different characteristic roughness, which were considered during the hydrodynamic modeling.

**Page 8 Lines 3-6: Comments about ks, it is not clear which comment refers to figure 4...**

Figure 4 (now Figure 5) shows the distribution of the values regarding the considered hydraulic parameters for each study site. The tests presented in the supplementary material are a complement to the figure. The analysis on the differences in roughness, depth and flow velocity can be done by looking only to Figure 4 (now Figure 5). Due to the great amount of data, differences between landscapes are not very noticeable in some cases, namely in water depths and flow velocities. Nevertheless, there are statistically significant different groups that are characterized by the different letters in the figure. Consequently, the authors decided to include in the supplementary material the tests results that support the figure construction and the author statements regarding the significant differences in the hydraulic characteristics of each riparian landscape. But these test do not show the exact same information as the figure. When we stated that those differences were statistically significant, we felt obliged to present the statistical tests that support our statements. Additional clarifications were included in the text.

Figure 5: It is not indicated to which reach refer each set of graphics...

This information was added to the figure caption which was changed according comments.

Page 8 Line 34: "The Eflow habitat consistently provides less habitat suitability during autumn...

This is an error. You should read "... nase juveniles and adults...". Text was corrected (now P12, L25).

Discussion: As I mentioned before, the modelling techniques that authors used have...

Limitations of the used techniques were discussed in the discussion section. The used terms were homogenized and the paragraph rewritten. P10, L19 was also rearranged (now P15, L3-7).

**Reviewer 2**

**Introduction**

**If I understand the argument correctly, the claim is that environmental flows for fish are...**

When the authors refer to species with longer lifecycles they are talking in general to mention that the usual approach on environmental flows only takes into consideration the intra-annual variability of the fluvial system but there is an inter-annual variation that can influence the life cycle of many organisms and which should be considered. In this case, the authors used riparian vegetation as the example for their case study. Riparian vegetation was then introduced in the next paragraph (page 2, lines 24 and 25), where the authors explain its connection to the flow regime and to the aquatic fauna when one mentions that riparian vegetation has a clear significance in the habitat improvement of aquatic systems, with references provided. Notwithstanding, riparian vegetation was mentioned in the text as example in the considered sentence (now P2, L20).

**Assuming that the paper focusses on fish and vegetation, the introduction should clarify much...**

Because of the manuscript length, particularly the introduction section, the authors did not find necessary to present a deep bibliographic review about the relations between riparian vegetation and aquatic fauna. The reason for this decision is that the main scope of the manuscript is on the efficiency of environmental flows when ignoring riparian requirements and fish species were used as a surrogate for the response of aquatic communities to the expected riparian landscape changes. This response was approached from an ecohydraulic point of view, in which the river hydraulics was the considered linkage between these two communities (page 2, lines 30-33). Even so, references were presented showing the influence of riparian communities on aquatic species assemblages in order to highlight the importance of restoring riparian vegetation not only for the improvement of these communities but also for the inherent improvements that such restoration can bring to aquatic species (page 2, line 26). Nevertheless, this paragraph was improved to better clarify how riparian vegetation and fish species relate to each other (now P2, L29-39).

**The research questions are not defined in a way that allows any other answer than that...**

If the authors understood correctly the concern of reviewer 2, the objective of this research is not to demonstrate that habitat modeling is the only good method for environmental flow assessment. In fact, the authors never make that statement throughout the manuscript. However, this study is based on habitat analysis and the authors needed to use habitat modeling to address their research questions. The term "overlook" of the second research question means the disregard that common approaches for environmental flows definition have for other biological communities. This comes from the bibliographic review in the introduction section where it is mentioned the need for environmental flows to address the ecological requirements of different biological communities rather than only a single biological group, which usually are always fish species. The question here is not if environmental flows disregarding riparian vegetation requirements allow for the degradation of these communities. We have already noticed that from previous studies. Based on the bibliographic review, the authors did not know the answer to question 1 but they raised their hypotheses. First, the authors question if the fish habitat would stay the same throughout the years facing the degradation of riparian vegetation due to flow regulation. If changes in habitat are noticeable, second question is: what is the extent of that change due to neglecting riparian requirements. Nonetheless, the authors understand the concern of the reviewer and this sentence was changed by a more consensual one (P3, L19-21).

**What is the "structural response" of riparian vegetation?**

Riparian vegetation is structured or arrayed in space and time along gradients in the three river dimensions: longitudinal, lateral and vertical. A response of riparian vegetation to a certain driver implying a change in this structure is denominated a structural response of riparian vegetation. This expression is widely used in vegetation science nomenclature (e.g. NRC, 2002; Naiman et al., 2005) and the authors provided such references on the text (now in P3, L23 and 24).

**Detailed remarks:**

**Page 1, line 33: would avoid using words like "truly".**

This word was changed by "Actually" (now P1, L33).

**Page 1, line 36: why "Therefore"?**

The authors mean that this is a consequence of the previous sentence. However, this was changed for a better understanding (now P1, L36).

**Page 2, line 6: "It is now in agreement" with only one reference is not very strong.**

The authors agree. More references were added, namely, Brisbane Declaration (2007); Arthington (2012); Poff et al. (1997); Acreman et al. (2014); Acreman and Ferguson (2010) and Davis and Hirji (2003). Now in P2, L8-10.

**Page 2, lines 10-11: what does "holistic" mean?**

There are different methodology types for environmental flow assessment. One of those types are named "holistic methodologies" (see Arthington et al., 2003; Dyson et al., 2003; Tharme, 2003; Arthington, 2012) also known as "function analysis" (see Dyson et al., 2003). Holistic methodologies are meant to address river systems as a whole. These methodologies emerged parallel in Australia and South Africa and share one same purpose: to protect or restore the flow-related biophysical components and ecological processes of the entire river system. An explanation and references were included in the text to support the context in which this term was mentioned (now in P2, L12-15).

**Page 2, line 15: why "clearly"?**

The authors state "clearly" supported in the systematic synthesis of the global literature regarding environmental flows done by Gillespie et al. (2014), which realized that the majority of the studies reported to fish response and given the importance of all trophic levels in sustaining freshwater ecological integrity, the predisposition towards monitoring of this traditional indicator taxa is a concern. According to this author, also Olden et al. (2014) found this tendency and therefore verified a clear need for diversification of monitoring strategies to cover less typically monitored taxa in future studies. Notwithstanding, this word "clearly" was removed without changing the meaning of the phrase.

**Page 2, line 16: do you need the word "biased"?**

This comes in line with the previous question about whether environmental flow assessments are prone to fish response evaluation rather than other biological groups. This word was removed.

**Page 2, line 35: "In what extent", does that exist?**

The authors propose to change to "to what extent".

**Methods**

**It is clear how the models link to the measurements. It is not clear at all how the models have...**

Additional information about models calibration was included in the text (now in P7, L24-29 and P8, L26-30). The calibration and validation of the models were referred to existing literature in order to control the length of the manuscript (see page 6, lines 3-6 and 19-20). Since the scope of the manuscript is not to present a particular model, the exhaustive description of the calibration methodology was not deemed necessary as plenty published papers already describe thoroughly the validation methodology of these models. Notwithstanding, the authors referred the calibration methodology employed in this case and presented the result of such calibration, so the reader can verify the accuracy of the model in this specific case study. Cohen's Kappa statistic was the chosen measure to evaluate the calibration of the CASiMiR-vegetation model because this is considered a good measure to analyze this model's accuracy and the most often used measure of inter-rater agreement for categorical classifications. Furthermore, it has an advantage over sensitivity, since it corrects the overall accuracy of model predictions by compensating for random agreement. Considering the River2D model, the authors first estimated the bed roughness coefficient, the roughness height, in accordance with the observations of bed material and bedform size for the natural flow regime. The final values of roughness height were obtained by calibrating the water surface elevation measured in different cross-sections in the field and the model results. For the different scenarios (i.e. Eflow regime and Eflow&Flush regime) the roughness height values were changed according to the expected riparian vegetation maps. For the roughness heights of the different vegetation types the authors supported their choice on expert judgment and literature. In the end, the employed models are widely used and scientifically accepted tools that were calibrated for the study sites according to recognized methodologies. Calibration results were analyzed by comparison to observed data and achieved a good classification according to different categorizations of map classification agreement. All of these provide confidence to the authors that the model results are right and simulate correctly the considered fluvial system. Furthermore, model uncertainty due to parameter estimation uncertainty can be performed by means of sensitivity analysis (Uusitalo et al., 2015). This was already assessed for both models by the authors and in both cases the models showed to be fairly robust to parameter input uncertainty (see Rivaes et al., 2013 and Boavida et al., 2013). These uncertainty analyses support the confidence of the authors that the uncertainty of models outputs are relatively small. The authors included a paragraph discussing this topic (P15, L20-24).

**Results**

**Detailed remarks:**

Done. All these remarks were addressed (see P11, L19-23, L27-28, L30-31).

**Discussion**

**Lines 23-26 on page 9 seem to be rather important. I would suggest that these could...**

Lines 23-26 were stated in a more prominent way. The two-way relation (back from fish to vegetation) is not considered in this modeling work. The two models employed in this study do not consider the effect of fish on vegetation or morphodynamics. The authors do not think this is applicable considering the river particle dimensions. River bed was considered stable during modeling runs (please see the response to reviewer 1 regarding this matter). The topics mentioned for this section by the reviewer 2 are pertinent and were discussed in the discussion section (now P13, L21-30 and P13, L34-37).

**Detailed remarks:**

**Page 9, line 16: what does "pushed through" mean?**

This means that such approach puts forward an ecological modeling procedure that is more realistic than the actual paradigm in the assessment of environmental flows by means of fish habitat modeling. This expression were changed by "enables" (now P13, L14).

**Page 9, line 36: why suddenly the term "substantially"?**

This means that the habitat availability originated by the Eflow changes a lot when compared to the natural and Eflow&Flush flow regimes. This expression was removed.

**Other remarks:**

**Please check the abstract. The second sentence is very difficult to understand...**

The second sentence of the abstract was modified for a better understanding.

The numbers mentioned in the abstract were introduced in a way that the authors thought to be more comprehensive and appealing to the reader without reading the entire article. Nevertheless, these specific numbers were introduced in the results section for a direct relation with the abstract values (now in P11, L36).

**The language needs to be improved. For example, several times the word "inputted" is...**

The manuscript was English revised by Elsevier Language Editing Services prior to the submission to HESSD journal and holds a certificate from this institution. The word "inputted" is the past tense and past participle of the verb "input". This can be found in different English dictionaries, like the Cambridge dictionary (http://dictionary.cambridge.org/dictionary/english/input) or the Oxford dictionary (https://en.oxforddictionaries.com/definition/input). If the reviewers and/or the handling editor still require a more thorough proofreading, the authors can readdress this issue to the Elsevier Language Editing Services in order to meet the expectations of the reviewers and handling editor.

Correspondence: Rui Rivaes (ruirivaes@isa.ulisboa.pt)

10 Abstract. Environmental flows remain biased toward the traditional biological group of fish species. AccordinglyConsequently, these flows ignore the inter-annual flow variability that rules species with longer life cycles and, thereby therefore -disregarding the long-term perspective of the riverine ecosystem. We analyzed the influence-importance of considering riparian requirements for the long-term efficiency of environmental flows. For that analysis, we modeled the riparian vegetation development for a decade facing

- 15 different environmental flows in two case studies. Next, we assessed the corresponding fish habitat availability of three common fish species in each of the resulting riparian landscapehabitat scenarios. Modeling results demonstrated that the environmental flows disregarding riparian vegetation requirements promoted riparian degradation, particularly vegetation encroachment. Such circumstance altered the hydraulic characteristics of the river channel where flow depths and velocities underwent local changes up to 10 cm and 40 cm s-1, respectively. Accordingly, after a decade of this flow regime, the available habitat
- area for the considered fish species experienced modifications in absolute from 18.16% up to 11009.75% when compared to the natural habitat. In turn, environmental flows regarding riparian vegetation requirements were able to maintain riparian vegetation near natural standards, thereby preserving the hydraulic characteristics of the river channel and sustaining the fish habitat close to the natural condition.

As a result, fish habitat availability never changed more than  $1\frac{76.17}{6}$  from the natural habitat.

**1 Introduction**

30

Freshwater ecosystems provide vital services for human existence but are on top of the world's most threatened ecosystems (Dudgeon et al., 2006; Revenga et al., 2000), primarily due to river damming (Allan and Castillo, 2007). The ability to provide sufficient water to ensure the functioning of freshwater ecosystems is an important concern as its capacity to provide goods and services is sustained by water-

- dependent ecological processes (Acreman, 2001). The relevance of this subject compelled the scientific community to appeal to all governments and water-related institutions across the globe to engage in environmental flow restoration and maintenance in every river (Brisbane Declaration, 2007). Actually Truly, this issue is a global reach topic, as all dams, weirs and levees change the magnitude of peak
- 35 flood flows of rivers to a certain extent (e.g., FitzHugh and Vogel, 2010; Maheshwari et al., 1995; Miller et al., 2013; Nilsson and Berggren, 2000; Uddin et al., 2014a, b). Therefore As a result of this, there are still

Comentado [RR1]: Modified to meet the specific comment of reviewer 1 on the title.

Comentado [RPGDRdS2]: Changed according to the specific comments of reviewer 1 regarding riparian vegetation modeling.

**Comentado [RR3]:** Modified according to detailed remarks of reviewer 2

**Comentado [RR4]:** Modified according to the detailed remarks of reviewer 2

[revised manuscript text omitted]

Formatado: Inglês (Estados Unidos) Comentado [RR6]: Included to address detailed remarks of reviewer 2

**Comentado [RR7]:** Included to address the comment of reviewer 2 about Introduction

**influence sediment surface characteristics and critical shear stress (e.g., Hassan et al., 2008; Statzner et al., 2003).**

Accordingly, Rriparian restoration is an indispensable implementation measure to recover the natural river processes and is the most promising restoration action in many degraded rivers (Palmer et al., 2014). Hence,

5 incorporating riparian vegetation requirements (the need for specific flows to preserve the naturalness of recruitment and meta-stability facing fluvial processes) into environmental flows could be an important contribution to fill in these gaps.

We have already noticed how environmental flow regimes disregarding riparian vegetation requirements allow for the degradation of riparian woodlands in the subsequent years following such river regulation

- 10 (e.g., Rivaes et al., 2015). However, we are not aware of studies assessing the comeback of this degradation again on the efficiency of those environmental flow regimes. The purpose of this study is to evaluate the effect of disregarding riparian vegetation requirements in the efficiency of environmental flow regimes regarding fish habitat availability in the long-term perspective of the fluvial ecosystem. We used an approach from an ecohydraulic point of view to evaluate the effects of riparian landscape degradation on
- 15 fish species By riparian landscape we mean the specific spatial patterns of riparian vegetation that result from ecological, geomorphological and hydrological processes, and are depicted by the existing patch mosaic with different vegetation types and succession phases. We were particularly interested in answering the following questions: i) are environmental flows exclusively addressing fish requirements capable of preserving the habitat availability of these aquatic species in the long-term? ii) If not, tolm what extent
- 20 caneould this overlook the disregard for riparian vegetation requirements derail the goals of environmental flows addressing only aquatic species as a result of the riparian landscapehabitat degradation? iii) Are environmental flows regarding riparian requirements able to maintain the habitat availability of fish species?

To approach these questions, we first modeled the structural response of riparian vegetation (please see Naiman et al., 2005 and NRC, 2002 for a better understanding about riparian vegetation structure) facing a

decade of different environmental flows in two different case studies. Next, we performed an assessment of habitat availability for fish species in each of the resulting riparian landscapehabitat scenarios. We are not aware of such a modeling approach ever being used in the appraisal of the long-term efficiency of environmental flow regimes, which can provide an extremely valuable insight of the expected long-term

30 effects of environmental flows in river ecosystems in advance.

**2 Methods**

35

**2.1 Study sites**

The two study sites were selected in the Ocreza River, East Portugal (Figure 1). This is a medium-sized stream that runs on schistose rocks for 94 km and drains a 1429 km2 watershed with a mean annual flow of 16.5 m3 s-1. The flow regime is typically Mediterranean (Gasith and Resh, 1999), with a low flow period interrupted by flash floods in winter (median of mean daily discharges in the winter months is 8.8 m3 s-1 and maximum annual discharges with a return period of 2, 5, 10 and 100 years are respectively 323, 549,

718 and 1314 m3 s-1) and a very low flow, even null at times, during summer (the first quartile and median

Comentado [RR8]: Text included according to the comment of reviewer 2 about Introduction

Formatado: Inglês (Estados Unidos)

Comentado [RR9]: Included to define the term "landscape", used to address the specific comment on Riparian vegetation modelling Page 6 Lines 12-13 of reviewer 1

Comentado [RR10]: Modified according to the comments of reviewer 2 about Introduction

**Comentado [RR11]:** Included according to the comment of reviewer 1 about Introduction

**Comentado [RR12]:** Included according to the specific comments of reviewer 1 on Methods

[revised manuscript text omitted]

**Comentado [RR15]: Text added to comply with the specific comments of reviewer 1 about Data Collection**

Comentado [RR16]: Included according to the comment of reviewer 1 Page 6 Line 8

Comentado [RPGDRdS17]: Text added to comply with the specific comments of reviewer 1 about Data Collection

flow at each sampling site. To develop Habitat Suitability Curves (HSC) for target fish size classes, microhabitat variables (flow depth, water velocity, dominant substrate and cover) were divided into classes, and histograms of frequencies of use and availability were constructed (Boavida et al., 2011). A summary on collected fish data, as well as data analysis to determine habitat use, availability and preference of fish

5 species regarding the considered analyzed variables, is provided as supplementary material (Appendix B – Table B2 and Figures B3 to B12).

**2.3 Flow regime definition**

Three flow regimes were considered for the modeling of riparian vegetation: i) the natural flow regime (hereafter named natural flow regime), ii) an environmental flow regime considering only fish requirements

- 10 (hereafter named Eflow regime) and iii) an environmental flow regime considering both fish and riparian requirements (hereafter named Eflow&Flush regime). The natural flow regime data was obtained from the Portuguese Water Resources National Information System (SNIRH, 2010). The environmental flow regimes used in this study are an adaptation from the environmental flow regime created by Ferreira et al. (2014) for the location of the study sites (Figure 2). These authors determined an environmental flow regime
- 15 presented in a multiannual fashion considering a decadal time frame and accounting for two different flow regime components: a monthly flow regime addressing fish requirements and a multiannual flow regime composed by floods with different recurrence intervals addressing riparian vegetation requirements. The first component, i.e., the flow regime addressing fish requirements (Eflow), was determined according to the Instream Flow Incremental Methodology (Bovee, 1982) and The flow regime addressing fish
- 20 requirements is was built on a monthly basis to embody the intra-annual variability ruling the main life cycle events of this biological group (Encina et al., 2006; Gasith and Resh, 1999), These mean monthly discharges addressing fish requirements that compose the Eflow aimed for the following goals: i) maximize the habitat of the target species while attributing the same weight for each species; ii) privilege the spawning months (spring; Santos et al., 2005) and promote the younger life stages during summer; iii) maintain the
- 25 characteristic intra-annual variability of the river flow; and iv) preserve the natural regime whenever the environmental flows suggest higher discharges. The second component of the environmental flow regime (floods with a certain recurrence interval) proposed by Ferreira et al. (2014) was determined according to Rivaes et al. (2015) and Likewise, the flushing flows of the riparian flow regime intend to characterize the inter-annual flow variability to which the arrangement of riparian vegetation communities respond
- 30 (Hughes, 1997). The flushing flows addressing riparian requirements in the Eflow&Flush regime were defined based on the need of riparian communities for the minimum necessary flushing flow regime to maintain the viability and sustainability of riparian vegetation, particularly, avoiding vegetation encroachment and conserving the ecological succession equilibrium of the riparian ecosystem (Rivaes et al., 2015). Therefore, the environmental flow regimes used in this study are considered an adaptation from
- 35 Ferreira et al. (2014) as we used just the fish-addressing component (only mean monthly discharges) as the standard procedure of an environmental flow regime considering only fish requirements (Eflow) and both components (mean monthly discharges and flushing flows) for the environmental flow regime addressing fish and riparian requirements (Eflow&Flush).

Comentado [RR18]: Included according to the comment of reviewer 1 Page 6 Line 8

Comentado [RR19]: Rearranged to address main concern 2 of reviewer 1

**Formatado: Inglês (Estados Unidos)**

Formatado: Inglês (Estados Unidos) Formatado: Inglês (Estados Unidos)

Formatado: Inglês (Estados Unidos)

**Formatado: Inglês (Estados Unidos) Formatado: Inglês (Estados Unidos) Formatado: Inglês (Estados Unidos)**

| Formatado: | Inglês (Estados Unidos) |
|------------|-------------------------|
| Formatado: | Inglês (Estados Unidos) |

**2.4 Riparian vegetation modeling**

The riparian vegetation modeling was performed using the *CASiMiR-vegetation* model (Benjankar et al., 2009). This tool simulates the succession dynamics of riparian vegetation, based on the existing relationships of the ecological relevant hydrological elements (Poff et al., 1997) and the vegetation metrics

- 5 that reflect riparian communities to such hydrological alterations (Merritt et al., 2010). The strengths of this model are the capacity of incorporating the past patch dynamics into every model run, the ability of working at a response guild level by using succession phases as modeling units, and the ability of providing the outputs in a spatially-explicit way. In turn, main disadvantages of this model can be attributed to the inexistence of a plant competition module or the lack of an incorporated hydrodynamic model.
- 10 The rational of this model is based on the fact that riparian communities respond to the hydrological and habitat variations on a time scale between the year and the decade (Frissell et al., 1986; Thorp et al., 2008), being that the flood pulse is the predominant factor on these population dynamics (Thoms and Parsons, 2002). For these reasons, the hydrological regime is inputted into the model in terms of maximum annual discharges as these discharges are considered as the annual threshold for riparian morphodynamic
- 15 disturbance that determine the succession or retrogression of vegetation. Notwithstanding, the model also predicts the annual riparian adjustments according to its vital rates in relation to groundwater depth, as well as the annual recruitment areas, based on the annual minimum mean daily discharges. The groundwater depth corresponding to the mean annual discharge of the river is also a model input used as a reference for the general habitat conditions that determine the expected riparian landscape according to the calibrated
- 20 thresholds of the riparian succession phases. Thus, the magnitude and duration of extreme low flows are accounted by CASiMiR-vegetation model. A complete detailing of model rational and parameterization can be found in Politti and Egger (2011) and Benjankar et al. (2011).

Model calibration was carried out in accordance with the methodology described in previous studies (García-Arias et al., 2013; Rivaes et al., 2013). Particularly, calibration was performed by running the

- 25 CASiMiR-vegetation model for a decade to simulate the effect of the local historic flow regime on riparian vegetation. The result of the model was then compared with an observed vegetation map that was surveyed in the same year of the one corresponding to the result of the model. This is an iterative process of trial an error where the parameter of shear stress resistance threshold of each succession phase is tuned to obtain the best calibration outcome (see Wainwright and Mulligan, 2004, for a better understanding). All the other
- 30 parameters, namely, patch age and height above water table ranges were determined based on the data collected in the field. This information is provided as supplementary material (Appendix A Table A5). During calibration, the riparian vegetation model achieved an agreement evaluation of 0.61 by the quadratic weighted kappa (Cohen, 1960), which is considered to be in good agreement with the observed riparian landscape (Altman, 1991; Viera and Garrett, 2005). This agreement evaluation can be understood as a
- 35 classification 61% better than what would be expected by a random assignment of classes. The riparian vegetation model was further validated in this specific watershed (Ferreira et al., 2014) with even better results (quadratic weighted kappa of 0.68). After calibration and validation (calibrated parameters provided as supplementary material; Appendix A Table A5), the riparian vegetation was modeled for periods of ten years according to the corresponding flow regimes (Table 1). Such modeling period was considered to
- 40 be long enough to avoid the influence of the initial vegetation conditions, while river morphological

Comentado [RR20]: Added according to the main concern 1 of the reviewer 1

**Comentado [RR21]:** Text added to meet the specific comments of reviewer 1 regarding riparian vegetation modeling Page 5 lines 36-39.

Comentado [RR22]: Included to address the comments of reviewer2 on Methods

**Comentado [RR23]:** Included according to the comment of reviewer 1 Page 6 Line 8

Comentado [RR24]: Included according to the comment of reviewer 1 Page 6 Line 8

changes still do not assume importance in vegetation development (Politti et al., 2014). Furthermore, during modeling, riverbed topography was considered fixed for several reasons: the study sites are located in a fairly steep valley in which river is not allowed to meander considerably during such a short time scale; the typical substrate of both study sites is armored and very coarse (boulders, large boulders and bedrock); in

- 5 these conditions the small monthly discharges intended to maintain aquatic fauna requirements are not able to create water depths and flow velocities capable of moving or eroding particles with the size of those found as substrate in the considered study sites (for a better understanding please see Alexander and Cooker, 2016; Clarke and Hansen, 1996; Hjulström, 1939); no significant differences were found during the substrate analysis of the different succession phases; prior knowledge of the authors show that the
- 10 considered floods do not bring noteworthy changes to river geomorphology during this period (Rivaes et al., 2015); the model calibration and validation results exhibited a good agreement with the observed riparian landscape while using the same methodology; by using a fixed topography it is possible to analyze the exclusive effect of riparian landscape degradation on the river hydraulics.

The resulting riparian vegetation maps were then used as the respective riparian landscapeshabitats+

15 (hereafter named natural, Eflow and Eflow&Flush landscapeshabitats) in the hydrodynamic modeling of fish habitat in each study site.

**2.5 Hydrodynamic modeling of fish habitat**

The hydrodynamic modeling was performed using a calibrated version of the River2D model (Steffler et al., 2002). This is a finite element model widely used in fluvial modeling studies for the assessment of habitat availability (Boavida et al., 2011; Jalón and Gortázar, 2007) that brings together a 2D hydrodynamic model and a habitat model to simulate the flow conditions of the river stretch and estimate its potential habitat value according to the fish habitat preferences. The strengths of this model are the fact of being public domain software and to be technically robust throughout a wide range of modeling circumstances.

- On the other hand, some limitations of this model are the non-incorporation of a morphodynamic module or the ability of embodying fuzzy logic rules during the computation of species habitat availability. The calibration procedure followed the methodology proposed by Boavida et al. (2013, 2015). Calibration was performed by iteratively adjusting the bed channel roughness to attain a good agreement of the simulated versus surveyed water surface elevations and velocity profiles in the surveyed cross-sections.
- 30 Boundary conditions were set according to the water surface elevations measured at the upstream and downstream cross-sections, and cCalibrated parameters are provided in supplementary material (Appendix A Tables A1, A2, A3 and A4).

The hydrodynamic modeling comprised the Eflow discharge ranges in the study sites  $(0 - 2 \text{ m}^3 \text{ s}^{-1} \text{ and } 0 - 5.5 \text{ m}^3 \text{ s}^{-1}$  for OCBA and OCPR, respectively) and was accomplished for each riparian landscape

35 scenariohabitat. The different riparian landscapes were represented in the hydrodynamic model by changing the channel roughness according to the spatial extent of the riparian succession phases, i.e., the channel roughness inputted to the model are the riparian landscape maps converted into channel roughness maps. Roughness is a critical feature influencing the physical variables of flow hydraulics (Chow, 1959; Curran and Hession, 2013), whose distinct combinations typify diverse functional habitats, which are selected by Comentado [RR25]: Added to address the reviewer 1 main concern 4. Formatado: Inglês (Estados Unidos)

Formatado: Normal

**Comentado [RR26]:** Added according to the main concern 1 of the reviewer 1

Comentado [RR27]: Included to address the comments of reviewer 2 on methods Comentado [RR28]: Included according to the comment of reviewer 1 Page 6 Line 8 fish according to its preference. The roughness classification of riparian vegetation succession phases was determined based on roughness measurement literature on similar vegetation types (Chow, 1959; Wu and Mao, 2007) and expert judgment during model calibration.

[revised manuscript text omitted]

**3 Results**

**3.1 Riparian vegetation modeling**

- 20 Different configurations of riparian habitat landscapes resulted from the riparian vegetation modeling according to the considered flow regimes in both case studies (Figure 3Figure 4). Nonetheless, the modeled response of riparian vegetation to each flow regime is similar in the two study sites. The riparian habitatlandscape, driven by the natural flow regime, presents a river channel that is largely devegetated, where Initial (IP) and Pioneer (PP) phases together represent approximately 43% and 35% of the study site
- 25 areas in OCBA and OCPR, respectively. In this habitatriparian landscape, Early Succession Woodland phase (ES) can only settle in approximately 8% of OCBA and 1% of OCPR areas. The floodplain succession phases, namely, Established Forest phase (EF) and Mature Forest phase (MF), represent nearly 40 and 10% of the study area for OCBA and, close to 42% and 23% for OCPR, respectively.
- -In contrast, the riparian habitat-landscape\_created by the Eflow regime is where the riparian vegetation
   and evolves toward mature phases due to the lack of the river flood disturbance. IP is now reduced to approximately 3% in OCBA and 6% in OCPR, while PP is inexistent in both cases. ES covers up to approximately 48% and 26% of the corresponding study areas, whereas EF and MF maintain about the same area in both case studies.
   -The riparian habitat-landscape\_driven by the Eflow&Flush regime shows the capacity of this flow regime
- 35 in hold back vegetation encroachment in both cases. In this habitatriparian landscape scenario, IP and PP are maintained at approximately 30% of the study site area in both case studies, whereas ES is kept under 21% in OCBA and only 2% in OCPR. Once again, EF and MF preserve their areas in both case studies.

Comentado [RR30]: Included according to the comment of reviewer 1 Page 6 Line 8

Comentado [RPGDRdS31]: Sub-sections added to comply with the reviewer 1 specific comments regarding the results Formatado: Título 2 -Summing up, the results of the riparian vegetation modeling show a riparian landscape degradation by vegetation encroachment in the Eflow landscape scenario when compared with the natural riparian landscape. Instead, the Eflow&Flush landscape scenario keeps approximately the same patch disposal and succession phasesphase's proportion as the natural landscape and therefore does not present significant

5 evidence offor riparian landscape degradation.

**3.2 Hydrodynamic modeling**

The changes undertaken by the riparian vegetation facing different flow regimes are able to modify the hydraulic characteristics of the river stretches (Figure 5Figure 4). Channel effective roughness heights ( $k_s$ ) change dramatically according to the considered riparian habitatslandscapes, increasing proportionally to

- 10 the encroachment level of vegetation in the study sites. In both case studies, the ks values of the Eflow habitats-landscape\_are clearly distinct and higher compared to the other two habitatsriparian landscapes (Figure 5). The ks values in the Eflow&Flush habitats-landscape were found to be between the values of Eflow and natural habitatslandscapes in the case of OCBA, and were very similar with the natural habitats landscape in the case of OCPR (Figure 5). Notwithstanding, in both case studies, the ks mean values are
- 15 statistical significantly different between all three habitats riparian landscapes (test results in supplementary material; Appendix C Table C1). The mean ks of the Eflow, Eflow&Flush and natural habitats-landscapes are 0.999, 0.709 and 0.462 m, respectively, in OCBA, and 1.034, 0.742 and 0.7178 m, respectively, in OCPR.

Changes also occur in flow depth and flow velocity for the considered discharge range of the proposed environmental flows (Figure 5). Although not so noticeable due to the great amount of data, differences are statistically significant. In OCBA, the Eflow habitat landscape creates a circumstance with statistically

- significant<del>ly</del> higher depths (mean depth is 0.402 m) and lower flow velocities (mean flow velocity is 0.128 m s-1) than the natural and Eflow&Flush habitats-landscapes. The t-tests on water depths (H0: true difference in means is equal to 0) revealed highly significant p-values (<0.001), respectively, for the
- 25 comparisons between Eflow and natural flow regimes, and Eflow and Eflow&Flush flow regimes. The ttests on flow velocities also derived a highly significant p-value (<0.001) in both the comparisons of natural versus Eflow regimes and Eflow versus Eflow&Flush flow regimes (test results in supplementary material; Appendix C – Tables C2 and C3). In contrast, depth and flow velocity are not significantly distinguishable between the natural and Eflow&Flush habitatslandscapes, where mean depth and flow velocity are 0.397
- 30 m and 0.136 m s-1, respectively, in the former, and 0.399 m and 0.135 m s-1 respectively, in the latter. -For the OCPR study site, flow depths are not significantly different (t-tests obtained p-values of 0.122 for natural versus Eflow regimes and 0.098 for Eflow versus Eflow&Flush flow regimes). (mMean values of flow depth for Eflow, Eflow&Flush and natural habitats-landscapes are 0.420, 0.417, 0.418, respectively.) butNonetheless flow velocities are different with statistical significance as the p-values of the t-tests for
- 35 natural versus Eflow and for Eflow versus Eflow&Flush were highly significant (<0.001), ; with tThe Eflow habitat-landscape createsing statistical significantly lower flow velocities (0.271 m s-1) when compared to the statistical significantly indistinct Eflow&Flush (0.277 m s-1) and natural (0.278 m s-1) habitats landscapes (test results in supplementary material; Appendix C Tables C2 and C3).

Formatado: Título 2

Comentado [RR32]: Included according to the comment of reviewer 1 Page 6 Line 8

Formatado: Inglês (Estados Unidos)

Comentado [RR33]: Included according to detailed remarks on Results by reviewer 2 Comentado [RR34]: Included according to the comment of reviewer 1 Page 6 Line 8

**Comentado [RR35]:** Included according to detailed remarks on Results by reviewer 2

Comentado [RR36]: Included according to detailed remarks on Results by reviewer 2

Comentado [RR37]: Included according to the comment of reviewer 1 Page 6 Line 8

Furthermore, when comparing water depths and flow velocities point by point, one can find differences between scenarios up to 10 cm in water depth and more than  $40 \text{ cm s}^{-1}$  in flow velocity. Accordingly, there are locations where the considered hydraulic parameters change considerably, shifting the habitat preference of fishes in one or two classes of the corresponding habitat preference curves.

5 In general, the Eflow landscapes present an increased channel roughness interfering with river flow and creating increased water depths and slower flow velocities when compared with the natural landscape. On the contrary, despite the increased channel roughness of the Eflow&Flush landscape, the water depths and flow velocities are very similar to the ones in the natural landscape. These results demonstrate that an environmental flow addressing exclusively fish requirements is not capable of preserving the habitat availability of the aquatic species for which was proposed in the long-term.

**3.3 Analysis of the Aaguatic habitat suitability for fish species analysis**

15

During a hydrological year, each riparian habitat-landscape provides different WUAs for the target fish species, with the same environmental flow regime addressing fish species (Figure 5Figure 6). Differences from the natural habitat suitability are greater in the Eflow habitat-landscape for both case studies. In OCBA, major differences in the WUA can be found almost all year round for the barbel juveniles, throughout autumn and winter months for the nase juveniles and during spring months for the calandino. Compared to the natural habitat-landscape, the WUA modifications instilled by the Eflow landscape habitat are on average approximately 12%, and are higher than 17% in a quarter of the cases reaching 80% in an extreme situation.

- Particularly, the Eflow landscapehabitat provides less habitat suitability during autumn and winter months for the barbel and nase juveniles, c. 17% and 14%, respectively. Likewise, in this habitatriparian landscape, the habitat suitability during spring months increases approximately 23% for the barbel juveniles and approximately 20 and 27% for the calandino juveniles and adults, respectively. On the other hand, throughout the year, the Eflow&Flush landscapehabitat provides a WUA very similar to the natural
- landscapehabitat.The habitat changes created by the Eflow&Flush landscapehabitat are on average25approximately 2% and never reach 8% for all species and life stages.

As for OCPR, major differences in WUA are seen almost all year round for calandino and nase, and exist particularly in spring months for barbel. WUA modifications due to the Eflow landscapehabitat are on average near 29%, being a quarter more than 50% and reaching up to more than 100% different in the most

- 30 extreme case. The Eflow landscapehabitat consistently provides less habitat suitability during autumn and winter months for the barbel and nase juveniles and adults, c. 50% and 38%, respectively, while the habitat suitability increases in approximately 46% of calandino. Moreover, the Eflow landscapehabitat provides an increased WUA during spring months in approximately 18% of the barbel adults and 71% of the calandino adults, while it decreases the habitat on average for approximately 7% of the remaining species and life
- 35 stages. Also in this case study, the Eflow&Flush landscapehabitat provides a WUA very similar to the natural landscapehabitat throughout the year. The habitat changes created by the Eflow&Flush landscapehabitat are on average near 3% and always less than 17% for all species and life stages. Accordingly, in both case studies, the WUA differences evidenced in the Eflow landscapehabitat revealed to be significant in several months by the χ2 test whereas this were never the case for the Eflow&Flush

| Formatado: Fonte: (Padrão) Times New Roman, 10 pt, Inglês
(Estados Unidos)               |
|---------------------------------------------------------------------------------------------|
| Formatado: Fonte: (Padrão) Times New Roman, 10 pt, Inglês
(Estados Unidos)               |
| Formatado: Fonte: (Padrão) Times New Roman, 10 pt, Inglês
(Estados Unidos)               |
| Comentado [RR38]: Included to address the other remarks
of reviewer 2 about the abstract |
|                                                                                             |

Comentado [RR39]: Included to address the specific comments of reviewer 1 about the results.

**Comentado [RR40]:** Included to address main concern 5 of reviewer 1.

Formatado: Título 2

Comentado [RR41]: Correction to address the specific comments about results Page 8 Line 34 of reviewer 1 landscapehabitats (test results provided in supplementary material; Appendix C – Tables C4, C5, C6 and C7).

The riparian-induced modifications on the WUAs are also confirmed by all the employed deviation measures (Table 2). According to RMSD, MAD and MAPD, the habitat provided by the Eflow landscapehabitat is always farther apart from the natural habitat for all species and life stages. In OCBA, the larger deviations occur for the barbel juveniles and nase adults, whereas in OCPR, the calandino adults and the barbel juveniles are the ones enduring greater habitat deviations from the natural circumstance, All together, these results reveal that the overlook of riparian requirements into environmental flows can derail the goals of environmental flows addressing only aquatic species by an extent of approximately an average

10 of 12 to 29% of the fish WUA's in the considered study sites as a result of the riparian habitatlandscape degradation. On the other hand, results reveal that environmental flows regarding riparian requirements are able to maintain the habitat availability of fish species as the WUA's in the study sites never change on average more the 3% in a decade.

Comentado [RR42]: Included according to the comment of reviewer 1 Page 6 Line 8

Comentado [RR43]: Included to address main concern 5 of reviewer 1.

**4 Discussion**

- 15 This study evaluated the benefits of incorporating riparian requirements into environmental flows by estimating the expected repercussions of riparian long-term changes driven by regulated flow regimes on the fish long-term habitat suitability. To this end, the riparian vegetation was modeled for 10-year periods according to three different simulated-flow regimes and results were inputted as the habitat basis for the hydrodynamic modeling and subsequent assessment of the fish habitat suitability in those riparian
- 20 landscapehabitats. Such ecological modeling approach, where a joint analysis is performed while embracing a suitable time response for the ecosystems involved, pushes throughenables a realistic biological-response modeling and substantiates the long-term research that is required in environmental flow science (Arthington, 2015; Petts, 2009). Furthermore, this approach allows one to foresee and assess the outcome of recommended flow regimes, which is an essential topic but has been poorly considered in
- 25 environmental flow science (Davies et al., 2013; Gippel, 2001). This research provides an insight of the expected long-term effects of environmental flows in river ecosystems, therefore unveiling the potential remarkable role of riparian vegetation on the support of environmental flows efficiency, which can transform the actual paradigm in environmental flow science.
- During modeling geomorphology was considered immutable and sediment transport originated by the environmental flow regimes was disregarded. River morphodynamics and its interactions with riparian vegetation constitute an important river process in many rivers, particularly in fine sediment rivers (e.g., Corenblit et al., 2009; Corenblit et al., 2011; Gurnell et al., 2012; Gurnell, 2014). However, the research on the temporal scales of geomorphic and ecological processes is still scarce in coarse-bed rivers (Corenblit et al., 2011), and simultaneously more complex and uncertain (Yasi et al., 2013). The error predictions from
- 35 best hydraulic predictors in this type of rivers can range between 50 to 200% (Van Rijn, 1993; Yasi et al., 2013). Disregarding such processes in these study sites was carefully considered. Given the above and the arguments mentioned in the methods section, we are confident that this option in this case will not bring tangible shortcomings to this research. Furthermore, the possible riverbed degradation effects due to the releasing of sediment-starving floods by the dam were not tested because according to our expert

**Comentado [RR44]:** Modified according to detailed remarks of reviewer 2 on Discussion.

**Comentado [RR45]:** Included to address the main concern 4 of reviewer 1 and comments on Discussion by reviewer 2

knowledge this will not pose a problem in this case. Such floods with similar recurrence intervals were already tested by Rivaes et al. (2015) in two river stretches of much smaller grain size (pebbles and sand) and results showed in both cases that such flood discharges were not relevant for riverbed degradation. The influence of fish species on geomorphology and riparian vegetation by ecosystem engineering, as it was

5 mentioned in the introduction, was not considered also during this study as it seemed fairly unrealistic in these case studies due to the general dimension of riverbed particles.
The results of the vegetation modeling illustrate how the natural flow regime generates morphodynamic

disturbances, without which the riparian vegetation is able to settle and age in the river channel. This is an important outcome that is essential to remember when providing environmental flow instructions.

- Subsequently, Consequently, of the latter, the mmicrohabitat analysis demonstrated that changes in the riparian landscapehabitat induce modifications in the hydraulic characteristics of the river stretches. The differences in mean values of these parameters are subtle between habitats-riparian landscapes but are statistically significant. Furthermore, Aa detailed analysis using a pairwise comparison of flow depths and velocities between scenarios show that modifications can reach 10 cm in water depth and more than 40 cm
- 15 s-1 in flow velocity in some places. The hydrodynamic modeling results show that the water flowing near the margins is more affected than the water flowing in deeper areas of the river channel. One reason for these results is certainly because this study is about the effects of riparian vegetation encroachment on the physical habitat due to the colonization of the river margins by woody riparian vegetation. Accordingly, there are locations where the considered hydraulic parameters change considerably, shifting
- 20 the habitat preference of fishes in one or two classes of the corresponding habitat preference curves. Such change can shift the habitat preference of fishes in one or two classes of the corresponding habitat preference curves. These changes are particularly important considering that an alteration of one class regarding these parameters is sufficient to change fish preferences from near null to maximum and vice-versa in many cases, as it can be seen in the preference curves provided in the supplementary material (Appendix B Figures B10, B11 and B12).
- 25 (Appendix B Figures B10, B11 and B12). The hydrodynamic modeling also indicated changes directly affecting the habitat suitability of the existing fish species according to the riparian landscapehabitat. Through time, the habitat riparian landscape shaped by the Eflow regime diverged substantially-in habitat suitability from the natural and Eflow&Flush landscapehabitats, and there were cases where the habitat suitability was modified by more than double.
- 30 The relationship between fish assemblages and habitat has long been acknowledged (e.g., Clark et al., 2008; Matthews, 1998; Pusey et al., 1993) and can have a significant impact on the ecological status and function of the existing fish communities (Freeman et al., 2001; Jones et al., 1996; Randall and Minns, 2000). Effectively, habitat loss is the major threat concerning fish population dynamics and biodiversity (Bunn and Arthington, 2002), thereby promoting population changes with a proportional response to the enforced
- 35 habitat change (Cowley, 2008). This is particularly true for the fish species considered in this study (Cabral et al., 2006). The habitat decrease for barbel and nase during autumn and winter months jeopardizes those species survival by refuge loss, which is particularly important in flashy rivers (Hershkovitz and Gasith, 2013), such as the Ocreza river and Mediterranean rivers in general. On the other hand, the habitat change during spring months undermines the spawning activity and consequently the sustainability of future
- 40 population stocks (Lobón-Cerviá and Fernandez-Delgado, 1984). The habitat increase of calandino during

Comentado [RR46]: Included to address the main concern 3 of reviewer 1

**Comentado [RR47]:** Included to address the comment of reviewer 2 about Discussion.

Comentado [RR48]: Included to address the comments of reviewer 2 about discussion.

Comentado [RR49]: Included according to the comment of reviewer 1 Page 6 Line 8

this period can be ecologically tricky due to the habitat plasticity of this species (Doadrio, 2011; Gomes-Ferreira et al., 2005), as well as its characteristic adoption for an r-selection strategy as an evolutionary response to frequently disturbed environments (Bernardo et al., 2003). Above all, one should not ignore that the relationships between fish assemblages and habitat are extremely complex (e.g., Diana et al., 2006;

5 Hubert and Rahel, 1989; Santos et al., 2011), being a consequence of the actual natural conditions (Poff and Allan, 1995; Poff et al., 1997) that when disrupted, may allow the expansion of more generalist and opportunistic fauna (Poff and Ward, 1989).

Our results indicate that environmental flows taking into account riparian vegetation requirements are able to preserve the naturalness of the riparian landscapehabitat and consequently, the maintenance of the fish

- 10 habitat suitability. Accordingly, the implementation of such measure measure in place of using environmental flows addressing only fish requirements can provide significant positive ecological effects in downstream reaches (Lorenz et al., 2013; Pusey and Arthington, 2003) and results in additional ecosystem services like stream bank stability, flood risk reduction or wildlife habitat (Berges, 2009; Blackwell and Maltby, 2006) while imposing minor revenue losses to dam managers (Rivaes et al., 2015).
- -The implementation of such environmental flows could provide an additional way to attain the "good ecological status" required by the Water Framework Directive (WFD). In addition, taking up a procedure such as this one can act both as 'win-win' and 'no-regret' adaptation measures during the second phase of the WFD, because it potentiates the improvement of other ecological indicators and mitigates the impacts of flow regulation, while being robust enough to account for different scenarios of climate change (EEA, 2005)
- Water science still lacks strong links between flow restoration and its ecological benefits (Miller et al.,

2012), particularly regarding long-term monitoring of environmental flow performance (King et al., 2015 and citations herein). Nevertheless, the outcomes of this study are a product of long-term simulations by models that were calibrated and validated for the corresponding watershed with local data in natural river

- 25 flow conditions. This standard procedure in modeling strengthens confidence in our predictions as the models proved to correctly replicate the response of the riparian and fish communities when paralleled with simultaneous observational data. In addition, model uncertainty due to estimation uncertainty in input parameters was previously assessed by means of sensitivity analyses on both models. In either case the models showed to be quite robust to the uncertainty of estimated parameter inputs (see Rivaes et al., 2013)
- 30 and Boavida et al., 2013) which reveal a relatively small uncertainty in the models outputs and provides additional confidence on the results.

35

40

In conclusion, we predict a change in fish habitat suitability according to the long-term structural adjustments that riparian landscapehabitats endure following river regulation. These changes can be attributed to the effects that altered riparian landscapehabitats have on the hydraulic characteristics of the river stretches. In our view, environmental flow regimes considering only the aquatic biota are expected to become obsolete in few years due to the alteration of the habitat premises in which they were based. This situation points to the unsustainability of these environmental flows in the long-term perspective of the fluvial ecosystem, failing to achieve the desired effects on aquatic communities to which those were proposed in the first place. An environmental flow regime that simultaneously considers riparian vegetation

**Comentado [RR50]:** Modified according to specific comments of reviewer 1 about discussion.

Comentado [RR51]: Included to address the concern of reviewer 2 about Methods

[revised manuscript text omitted]

---

## Author Response (AR2)

Dear Handling Editor Stan Schymanski,

On behalf of all authors, I would like to thank you, the anonymous reviewer #1 and the reviewer #2 Maurits Ertsen, for the acceptance of this article with corrections. We are sure that such interactive review led to a great improvement of the first draft of this manuscript.

Concerning the corrections proposed by the reviewer #2 Maurits Ertsen we must say that for page 1, line 34, this is not a mistake and we indeed meant "reach" and not "research". With this sentence, we want to say that environmental flow restoration is in fact a topic that is pertinent to the majority of the rivers worldwide and not just for a particular type of rivers, continent or climate.

As for the second suggestion of reviewer #2 Maurits Ertsen, we changed the word "comeback" by "return effect" which we think will meet the preference of this reviewer.

Once again, thank you for allowing us the opportunity to revise this manuscript and contribute to Hydrology and Earth System Sciences Journal. Please do not hesitate to contact me if you have any further question.

Best regards,

Rui Pedro Rivaes